# WHEN DISTANCE DISTRACTS: REPRESENTATION DISTANCE BIAS IN BT-LOSS FOR REWARD MODELS

## ABSTRACT

Reward models are central to Large Language Model (LLM) alignment within the framework of RLHF. The standard objective used in reward modeling is the Bradley-Terry (BT) loss, which learns from pairwise data consisting of a pair of *chosen* and *rejected* responses. In this work, we analyze the per-sample gradient of BT-loss and show that its norm scales with two distinct components: (1). the difference in predicted rewards between chosen and rejected responses, which reflects the **prediction error**, and critically, (2). **representation distance** between the pair measured in the output space of the final layer. While the first term captures the intended training signal, we show that the second term can significantly impact the update magnitude and misalign learning. Specifically, pairs with small representation distance often receive vanishingly weak updates, even when misranked, while pairs with large distance receive disproportionately strong updates. This leads to gradients from large-distance pairs to overshadow those from small-distance pairs, where fine-grained distinctions are especially important. To overcome this limitation, we propose NORMBT, an adaptive pair-wise normalization scheme that balances representation-driven effects and focuses learning signals on prediction error. NORMBT is a lightweight, drop-in integration to BT loss with negligible overhead. Across various LLM backbones and datasets, NORMBT improves reward model performance consistently, with notable gains of over $5\%$ on the Reasoning category of RewardBench, which contains numerous small-distance pairs. This work reveals a key limitation in the widely used BT objective and provides a simple, effective correction.

## 1 INTRODUCTION

Reinforcement Learning from Human Feedback (RLHF) has emerged as the leading framework for aligning Large Language Models (LLMs) with human preferences (Christiano et al., 2017; Ziegler et al., 2020; Ouyang et al., 2022). Instead of traditional supervision, RLHF uses preference signals to guide LLMs in applying their vast pretrained knowledge more efficiently and aligned with human intents. This paradigm has become the central alignment pipeline, driving both research and large-scale deployment. RLHF involves two main stages, with the reward model (RM) serving as a proxy for human preferences. In the first stage, the RM is trained on pairwise preference data, learning to predict the quality and alignment of candidate responses with human judgment. In the second stage, this RM scores outputs from the target policy LLM, providing optimization signals for reinforcement learning. Therefore, the quality of the reward model directly governs the effectiveness of learning for the policy LLM.

Reward modeling is inherently challenging. Human preference data is often ambiguous and noisy(Liu et al., 2025; 2024c; Cui et al., 2023). Furthermore, it tends to span a wide range of tasks with varying alignment goals. For instance, in safety-oriented scenarios, the preferred response may be refusing to answer harmful queries. In contrast, reasoning-heavy tasks such as coding depend on precise, step-by-step correctness. A reward model must therefore learn to faithfully capture these varied notions of "good" behavior across diverse data. To achieve this, it must extract equally reliable learning signals across both broad, easy-to-separate cases and fine-grained, subtle distinctions. This makes the design of the training objective and its update dynamics especially important, since even minor bias in how models learn across different types of data can undermine the overall alignment process.

Figure 1: An illustration where a pair (top) receives inherently large update due to large representation distance, and a pair (bottom) receive weak update due to small distance

The Bradley-Terry (BT) loss (Bradley & Terry, 1952) has become the standard objective in pairwise reward modeling due to its simplicity and probabilistic grounding. Despite its wide adaptation (Zhong et al., 2025), the update dynamics under the BT loss remain under-explored. As in most supervised learning settings, we expect the magnitude of reward model update to be driven by prediction error, i.e., how wrong the model is in ranking chosen versus rejected responses. When the model misranks a pair by a large margin, updates should be strong to correct the error; when the predicted ranking is correct, updates should be weak to avoid unnecessary shifts.

However, our analysis shows that this is not the case under the BT objective. By decomposing the BT gradient, we identify two factors that determine update size: (1) prediction error, calculated as the difference between the true rank and the predicted rank, and critically, (2) the representation distance between the pair of chosen and rejected responses. This coupling means that regardless of prediction accuracy, small-distance pairs tend to receive weak updates, while highly dissimilar pairs can trigger large updates even when correctly ranked. This behavior contradicts typical intuition, introducing spurious signals that undermine learning efficiency.

In fact, the diversity of preference data highlights this issue in how the training loss translates pairwise comparisons into learning signals. Figure 1 illustrates this issue with two examples. In example 1, drawn from Safety task, the chosen response refuses a harmful query while the rejected response attempts to answer it. In example 2, from Reasoning task, the chosen response provides the correct coding solution while the rejected response contains a logical error that renders it entirely wrong. Both pairs reflect a clear notion of "good" versus "bad," yet they differ significantly in representation distance: the safety pair is far apart in representation space, while the reasoning pair is close. Under the BT loss, such cases become problematic. Rather than true prediction error, representation distance disproportionately dictates the learning signal. Figure 2 shows that this effect extends beyond isolated cases and is visible across datasets.

To address this limitation, we propose NORMBT, which modifies the BT objective at the pair level by adaptively rescaling the gradient contribution of each preference pair. The raw BT update is driven by two parts: prediction error and representation distance. We show that the latter component can suboptimally inflate or suppress update size. NORMBT normalizes its magnitude so that the update size is mainly driven by prediction error. This modification ensures the intuition in parameter updates, that model performance determines the strength of learning signals. NORMBT is designed as a simple and direct integration into BT loss. It preserves the probabilistic foundation, requires no architectural changes, and incurs negligible computational overhead. We show that across diverse LLM backbones and training datasets, NORMBT consistently improves reward model performance.

Our contributions are summarized as follows:

1. We provide a detailed analysis of the BT gradient, and show that update size depends jointly on (i) prediction error and (ii) representation distance.
2. We demonstrate that this coupling introduces spurious learning signals: small-distant pairs receive inherently weak updates even when misranked, while distant pairs receive strong updates. This undermines the model's learning from fine-grained distinctions.
3. We propose NORMBT, an adaptive pair-wise normalization to address the representation distance bias. We show that NORMBT consistently improve model performance across different base models and datasets. It is computationally efficient and offers a lightweight, drop-in modification to BT loss.

## 2 METHOD

In this section, we first decompose the gradient of BT-loss, and establish its connection to representation distance between response pairs. We provide evidence to illustrate how this leads to biased updates, and finally introduce NORMBT as proposed solution.

### 2.1 GRADIENT NORM ANALYSIS

Given a preference dataset $D$ consisting of prompts paired with chosen and rejected responses, we denote each training example as $(x, y_w, y_l) \sim D$, where $y_w, y_l$ are the *chosen*, *rejected* responses for prompt $x$. The reward model $r_\theta$ parameterized by $\theta$ assigns a scalar score to each $(x, y)$ pair.

The Bradley-Terry (BT) loss is defined as:

$$\mathcal{L}_{\text{BT}}(\theta) = -\mathbb{E}_{(x,y_w,y_l)\sim D} \left[ \log \sigma \big( r_\theta(x, y_w) - r_\theta(x, y_l) \big) \right],\tag{1}$$

where $\sigma(\cdot)$ is the sigmoid function. This objective maximizes the likelihood that $r_\theta$ assigns a higher reward for $y_w$ than for $y_l$.

For clarity, we abbreviate $r_w = r_\theta(x, y_w), r_l = r_\theta(x, y_l)$, and the difference in rewards $d = r_w - r_l$. Under this notation, the gradient of equation 1 for a pair $(y_w, y_l)$ takes the form

$$\nabla_\theta \mathcal{L}_{\text{BT}} = \big( \sigma(d) - 1 \big) \nabla_\theta (r_w - r_l).\tag{2}$$

The gradient norm $\|\nabla_\theta \mathcal{L}\|$ directly reflects how much the model is updated to correct the error, which we now decompose.

**Gradient Norm.** Consider the standard reward modeling parametrization $\theta = (\phi, \mathbf{w}_s)$, with a linear score layer $(\mathbf{w}_s^T h + b_s)$ attached to an LLM backbone $\phi$. Let $h_\phi(\cdot)$ denotes the final-layer representation output from $\phi$, then the reward is given by

$$r_\theta(x, y) = \mathbf{w}_s^T h_\phi(x, y) + b_s.\tag{3}$$

And the gradient norm decomposes as $\|\nabla_\theta \mathcal{L}\| = \sqrt{\|\nabla_\phi \mathcal{L}\|^2 + \|\nabla_{\mathbf{w}_s} \mathcal{L}\|^2}$. We first consider the component of $\phi$. Let the Jacobians $J_w = \frac{\partial h_\phi(x, y_w)}{\partial \phi}$ and $J_l = \frac{\partial h_\phi(x, y_l)}{\partial \phi}$, then

$$\nabla_\phi \mathcal{L} = \big( \sigma(d) - 1 \big) \big( J_w - J_l \big)^\top \mathbf{w}_s.\tag{4}$$

If the embedding map is locally $L_g$-Lipschitz-smooth, $\|J_w - J_l\| \leq L_g \|h_w - h_l\|$, we have

$$\begin{aligned}\|\nabla_\phi \mathcal{L}\| &= \big| \sigma(d) - 1 \big| \cdot \|\mathbf{w}_s\| \|(J_w - J_l)^\top\| \\ &\leq \big| \sigma(d) - 1 \big| \|\mathbf{w}_s\| \cdot L_g \|h_w - h_l\|.\end{aligned}\tag{5}$$

Secondly, at the linear score layer $\mathbf{w}_s$,

$$\|\nabla_{\mathbf{w}_s} \mathcal{L}\| = \big| \sigma(d) - 1 \big| \cdot \|h_w - h_l\|.\tag{6}$$

Summing both contributions from $\phi, \mathbf{w}_s$, we have the following expression deriving from

$$\begin{aligned}\|\nabla_\theta \mathcal{L}\| &= \big| \sigma(d) - 1 \big| \cdot \big\| \nabla_\theta (r_w - r_l) \big\| \\ &\leq \big| \sigma(d) - 1 \big| \cdot \sqrt{\big( \|\mathbf{w}_s\| \cdot L_g \|h_w - h_l\| \big)^2 + \|h_w - h_l\|^2} \\ &= \underbrace{\big| \sigma(d) - 1 \big|}_{\text{prediction error}} \cdot \underbrace{\big( k \|h_w - h_l\| \big)}_{\text{representation distance}},\end{aligned}\tag{7}$$

where $k = \sqrt{1 + \big( L_g \|\mathbf{w}_s\| \big)^2}$.

This decomposition makes clear that the strength of update depends jointly on (1). **prediction error**, or how well the model already ranks the responses as captured by the reward difference $d$, and (2). model's sensitivity to the pair, which in practice is governed by **representation distance**.

**Motivations.** The term $\nabla_\theta(r_w - r_l)$ reflects how to adjust $\theta$ so that the chosen reward $r_w$ increases while the rejected reward $r_l$ decreases. It captures the *sensitivity of model parameters to this pair*. As we have shown through decomposing into backbone and score layer contributions in Eq. 7, this sensitivity is closely related to the representation distance $\|h_w - h_l\|$. When representations distance between $(y_w, y_l)$ is small, their gradients nearly cancel, producing only a small update. When the representations are far apart, the gradient difference is larger, giving a stronger update.

To better illustrate this decomposition, Figure 1 presents two representative examples from preference data. In the first example, $(y_w, y_l)_{\text{Safety}}$ are dissimilar responses, yielding a large representation distance term $\|\nabla_\theta(r_w - r_l)\|$. In the second example, the two responses differ by a logical error, making them nearly identical in form and thus close in representation distance. Thus the scale factor in update magnitude of Eq. 7 is naturally larger for $(y_w, y_l)_{\text{Safety}}$ than that for $(y_w, y_l)_{\text{Reason}}$. Appendix D shows additional examples drawn from evaluations.

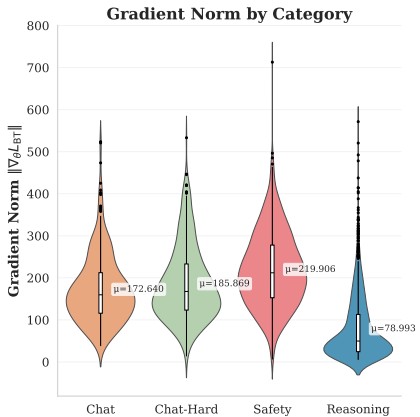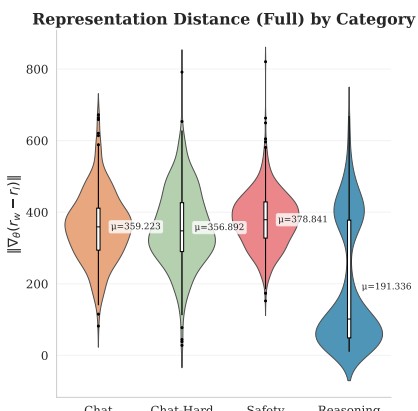

Figure 2: **Gradient Information Across Dataset.** Gradient norms (thus update sizes) vary widely by task under the BT-loss, corresponding to the variation in representation distance on the right. In particular, Reasoning pairs exhibit the smallest distance and correspondingly the weakest updates.

**Q1: Does the disparity in representation distance appear across dataset?**

This effect is not limited to isolated cases but extends across datasets. We illustrate this using RewardBench (Lambert et al., 2024), a benchmark for reward models that provides categorized preference data (Chat, Chat-Hard, Safety, and Reasoning), allowing easy understanding for cross-task comparisons. Figure 2 compares the distribution of gradient norms (left) and representation distance $\|\nabla_\theta(r_w - r_l)\|$ (right) across pairs in RewardBench. The left panel shows that the gradient update sizes under the BT objective vary widely by task, with the Reasoning category exhibiting the *smallest average gradient norm*, despite being a domain where fine-grained distinctions are crucial. The right panel reveals why: Reasoning pairs also have the *smallest representation distance* on average, whereas Safety and Chat pairs are much farther apart.

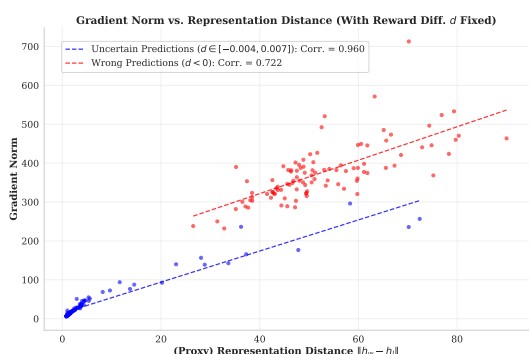

Figure 3: Update sizes at fixed reward difference.

This observation directly supports our intuition. Since BT-loss couples prediction error with representation distance (Eq. 7), updates for small-distance pairs are inherently weaker. While such small-distance, weak-update cases appear frequently in reasoning tasks under this specific data distribution, the underlying issue is more general: it arises from the representation distance, rather than any single task domain.

**Q2: What happens if representation distance outweighs prediction error?**

Figure 3 provides a perspective showing how representation distance can dominate, by comparing pairs with similar prediction errors as captured by the reward difference $d$. Even within each bin, which holds prediction error nearly fixed, update size grows with representation distance.

In preference learning, minor differences can decisively determine correctness. For example, a single logical mistake in reasoning tasks renders an otherwise valid solution entirely wrong. If such pairs inherently receive only weak updates due to small representation distance, the model under-corrects clear misrankings and fails to separate chosen from rejected behavior. Conversely, when certain pairs can trigger large updates regardless of accuracy, the reward model may overemphasize superficial differences. We argue that update strength should be more aligned with true prediction errors. When representation distance dominates, the learning signal may be misaligned or misleading for optimization.

## 2.2 SAMPLE-WISE NORMALIZATION

To align BT update sizes more directly with prediction error, we introduce into the loss function a detached per-pair weight $w_i(x, y_w, y_l)$ that scales inversely with the norm of representation distance. In principle, this weight can be defined as

$$w_i = \frac{1}{\|\nabla_\theta(r_w - r_l)\|}. \tag{8}$$

Thus the gradient norm simplifies to

$$\left\|\nabla_\theta\hat{\mathcal{L}}\right\| = w_i\left|\sigma(d) - 1\right|\left\|\nabla_\theta(r_w - r_l)\right\| = \left|\sigma(d) - 1\right|. \tag{9}$$

As $w_i$ directly removes the multiplicative factor from the representation distance, the update size does not depend on this magnitude.

**Representation Distance Proxy.** However, computing $w_i$ is impractical for large datasets, as it requires additional backward operations to obtain per-sample gradient information. Given the established connection $\|\nabla_\theta(r_w - r_l)\| \propto \|h_w - h_l\|$ in Sec 2.1, we consider $\|h_w - h_l\|$ as a direct proxy, and define the pair-wise weight

$$\tilde{w}_i = \frac{1}{\|h_w - h_l\|}, \tag{10}$$

This proxy offers three key advantages. First, it is **simple and efficient**: $(h_w - h_l)$ can be obtained conveniently during the forward pass without additional computations. Second, it is **empirically justified**: beyond theoretical derivations in Eq. 7, we provide empirical evidence that this proxy is well-correlated with the full representation distance $\|\nabla_\theta(r_w - r_l)\|$ in Appendix E. Importantly, this proxy is not just approximate but exact at the linear score head (Eq. 6). Since the score head directly controls the reward prediction, it is also natural to view its gradient as a convenient normalization signal.

Finally, the use of $\|h_w - h_l\|$ admits an **alternative, simple interpretation**. As this proxy corresponds to the final layer embedding distance between the response pair, it provides an intuitive viewpoint. Pairs with small embedding differences often correlate with responses that are e.g., close in form, structure, semantics, etc, such as the Reasoning example shown in Figure 1. From our analysis, their updates are inherently weak under the BT objective. Normalizing by $\frac{1}{\|(h_w - h_l)\|}$ compensates for this and upweights such pairs, allowing the training signal to allocate more attention to fine-grained distinctions. Conversely, highly dissimilar responses naturally receive large updates and are thus down-weighted, reducing the tendency to overemphasize superficial differences in representation space.

**Stabilizing Scale Drift.** Since the scale of embeddings can vary during training, raw values of $\tilde{w}$ can drift and tighten or loosen normalization inadvertently (Appendix F). To make the weighting $\tilde{w}$ invariant to such scale drift, we use an exponential moving average (EMA) of the batch statistics.

Under this formulation, we do not eliminate the magnitude of the representation-distance component entirely, but instead rescale each pair relative to the running mean. Let $\mu_t$ denote the tracked EMA

estimates of the mean embedding difference at optimization step $t$. We define the normalization weight for a training pair $(y_w, y_l)$ relative to $\mu_t$,

$$\tilde{w}_t(y_w, y_l) = \mu_t / (\|h_w - h_l\| + \epsilon) \tag{11}$$

where $\epsilon$ is a small constant for numerical stability.

Let $\hat{\mu}_t = \mathbb{E}_{\text{batch,t}}[\|h_w - h_l\|]$ denote the mean representation difference in the current batch, the EMA estimate is then updated as

$$\mu_{t+1} \leftarrow \beta \mu_t + (1 - \beta)\hat{\mu}_t, \tag{12}$$

with $\beta = 0.9$ by default. Using the EMA-relative scaling $\mu_t$ (rather than a fixed constant) adapts the normalization strength over time while smoothing batch noise, giving weights that are stable and adaptive to distributional shifts. Regardless of the backbone's representation scale drifts, $\tilde{w}$ stays near unity on average and the effective loss scale remains stable.

**Final Objective.** Concretely, the NORMBT objective incorporates both the proxy for representation distance and EMA-based stabilization, given by the following

$$\mathcal{L}_{\text{NormBT}}(\theta) = -\mathbb{E}_{(x,y_w,y_l)\sim D} \left[ \tilde{w}(y_w, y_l) \cdot \log \sigma(r_w - r_l) \right]. \tag{13}$$

In effect, NORMBT rescales the gradient contribution of training pairs such that small-distance pairs are not overshadowed by large-distance pairs.

## 3 EXPERIMENTS

### 3.1 SETUP

**Base Models.** Following standard practice in reward modeling as sequence classifier, we attach a linear score head to a pretrained LLM backbone to output scalar rewards. We experiment with two open-source base models: **gemma-2b-it** (Team et al., 2024) and **Llama-3.2-3B-Instruct**[1]. These choices evaluate NORMBT across distinct backbone families and capacities.

**Datasets.** We train the reward models on two pairwise preference datasets, yielding a total of four experimental settings (two backbones × two dataset): (1). **Unified-Feedback**[2]**.** This is a large and diverse collection of pairwise feedback aggregated from multiple open-source datasets. We train on a random subset of 80K preference pairs from Unified-Feedback. (2). **Skywork-Reward-Preference-80K-v0.2** (Liu et al., 2024a). This is a curated dataset of 80K high-quality preference pairs covering diverse capability and knowledge domains. It is the decontaminated version from Skywork-Reward-Preference-80K-v0.1 (Liu et al., 2024a) that removes overlaps with the evaluation benchmark RewardBench, ensuring no contamination between training and evaluation.

**Baselines.** We compare the performance of NORMBT with several loss objectives, including (1) *BT Baseline* the standard Bradley–Terry loss in Eq 1; (2) *BT + Margin* (Touvron et al., 2023) which incorporates the ground-truth reward margin $m$ into BT-loss; (3) *BT + Margin (outside)* (Wang et al., 2025) which varies by introducing $m$ as a scale factor to loss; and (4) *BT + label smoothing* (Liu et al., 2024a) which is a widely used regularization where soft labels replace hard binary targets. The two margin-based variants explicitly modify the prediction-error term by injecting ground-truth reward differences, thereby adjusting update strength according to external signals. Label smoothing also alters the prediction-error term, but by uniformly reducing its magnitude. These methods serve as natural comparisons since they manipulate the error strength, whereas our approach targets representation distance.

For all BT baselines, we conduct an extensive grid search over learning rates and report the best-tuned results. For BT-based variants (margin, margin-out, label smoothing), we search over a range of learning rates centered around the best value found for BT and report the best configuration. Formulation for each baseline and additional training details are included in Appendix B.

---

[1]https://huggingface.co/meta-llama/Llama-3.2-3B-Instruct
[2]https://huggingface.co/datasets/llm-blender/Unified-Feedback

Table 1: Results on RewardBench from four settings (two base models × two datasets)

(a) Reward models trained from Unified-Feedback (80K)

| Reward model | Average | Chat | Chat-Hard | Safety | Reasoning |
|---|---|---|---|---|---|
| Base Model: gemma-2b-it | | | | | |
| BT (baseline) | 72.25 | 95.25 | 40.35 | 77.97 | 75.41 |
| BT + margin | 71.23 | 95.81 | 37.50 | 78.65 | 72.98 |
| BT + margin out | 72.53 | 96.09 | 38.38 | 77.57 | 78.09 |
| BT + label smooth | 69.95 | 93.85 | 37.72 | 75.95 | 72.28 |
| NORMBT (ours) | 73.57 | 95.81 | 39.80 | 77.97 | 80.71 |
| Base Model: Llama-3.2-3B-Instruct | | | | | |
| BT (baseline) | 75.24 | 95.53 | 49.89 | 81.69 | 71.70 |
| BT + margin | 74.15 | 97.77 | 47.59 | 81.28 | 69.97 |
| BT + margin out | 74.84 | 96.65 | 45.83 | 84.05 | 72.81 |
| BT + label smooth | 74.39 | 95.53 | 49.78 | 79.73 | 72.52 |
| NORMBT (ours) | 76.96 | 96.93 | 49.78 | 84.19 | 76.93 |

(b) Reward models trained from Skywork-Reward-Preference-80K-v0.2

| Reward model | Average | Chat | Chat-Hard | Safety | Reasoning |
|---|---|---|---|---|---|
| Base Model: gemma-2b-it | | | | | |
| BT (baseline) | 78.63 | 81.28 | 73.36 | 82.43 | 77.46 |
| NORMBT (ours) | 80.12 | 83.80 | 73.46 | 82.50 | 80.71 |
| Base Model: Llama-3.2-3B-Instruct | | | | | |
| BT (baseline) | 80.31 | 86.03 | 78.29 | 89.86 | 67.05 |
| NORMBT (ours) | 81.48 | 83.80 | 78.73 | 88.78 | 74.60 |

## 3.2 MAIN RESULTS

Table 1a reports the performance of all reward models trained on Unified-Feedback (80K), on both base models. And Table 1b presents the results trained on Skywork-Reward-Preference-80K-v0.2. Across all four settings, the NORMBT objective consistently outperforms the BT baseline, showing that mitigating representation-distance bias improves reward modeling.

**NORMBT vs. BT.** In particular, strong performance gains are observed in the Reasoning category, where NORMBT improves accuracy by more than $5\%$ on average. This is consistent with our intuition: preference pairs in the Reasoning category from RewardBench tend to have smaller representation distances (as shown in Figure 2). Under the vanilla BT objective, these pairs receive disproportionately weak updates, even when misranked. By removing this representation-driven effect in parameter updates, NORMBT provides stronger updates that are more aligned with prediction errors, leading to the largest performance improvement amongst these pairs.

**NORMBT vs. BT-variants.** The margin-based baselines (BT+Margin, BT+Margin out) do not yield consistent improvements over the vanilla BT loss. Although these methods introduce ground-truth margins to directly supervise the strength of the prediction-error term, they do not correct for the scaling bias introduced by representation distance. Cases where margin-based training hurts performance suggest possible overfitting to noisy margin annotations. This highlights that simply enriching supervision with ground-truth requires high-quality annotations, and is still insufficient to address the structural bias of BT updates. In contrast, NORMBT requires no external signals beyond the standard pairwise preference data, and outperforms both variants of margin-based baselines. This shows that addressing the representation bias directly is more reliable and broadly applicable.

Lastly, label-smoothing does not improve over the vanilla BT objective. As shown in Eq 18, label-smoothing softens the target distribution by uniformly reducing the prediction-error magnitude in the gradient. While this can help calibration, it also weakens the update strength globally, including

pairs with small-representation distance where the gradient is already small. The results further align with this intuition, as the largest drop in performance is reflected in the Reasoning category (e.g., from 75.41 to 72.28 in gemma-2b-it with Unified-Feedback).

**Analysis.** Beyond the results presented in Table 1, we analyze *where* the performance gains of NORMBT originate. Figure 4 shows the subset of RewardBench pairs where the BT Baseline and NORMBT predictions disagree. That is, cases where one model is correct while the other is wrong. The x-axis bins response pairs by their representation distance $\|h_w - h_l\|$, where $h$ are derived from the base model gemma-2b-it, and the y-axis shows the count of these exclusive correct predictions. For ease of interpretation, we divide the range into small, medium, and large distances roughly based on EMA statistics of the embedding distribution; these divisions are intended only for discussion.

The clearest gains of NORMBT over BT appear in the **small-distance regime**. For pairs in this range, BT updates are weak and leave misrankings or ambiguous predictions insufficiently corrected. NORMBT resolves this by normalizing the representation distance factor to an average baseline, ensuring prediction error drives the update. Accordingly, the histogram shows a clear concentration of additional correct predictions from NORMBT in this region, in line with our intuition. Appendix D shows examples from this region that NORMBT predicts correctly while BT doesn't.

In the **medium- and large-distance regimes**, results differ slightly across datasets. With Unified-Feedback, NORMBT still yields general improvements, while on SKywork-80K both methods perform comparably. This suggests that when the representation distance is large, BT already produces strong updates, and the representation-distance term is less of a bottleneck. By down-weighting pairs from the large-distance range, NORMBT maintains parity or modest gains, suggesting that improvements on small-distance pairs do not come at the expense elsewhere.

Overall, these results show that NORMBT provides the largest performance gains on pairs with small representation distance, while remaining comparable across other distance ranges.

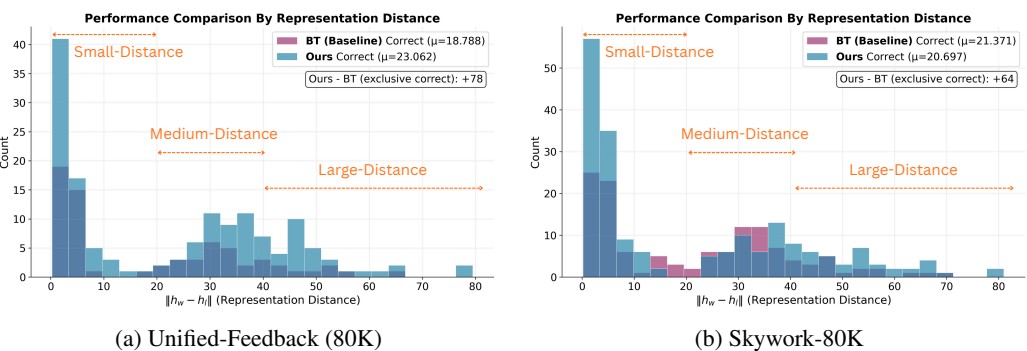

(a) Unified-Feedback (80K)  (b) Skywork-80K

Figure 4: Comparison on RewardBench pairs where two models disagree. Pairs are binned by representation distance $\|h_w - h_l\|$ computed from gemma-2B-it. The largest gains for NORMBT appear in the small-distance regime, consistent with our analysis that BT under-updates such pairs.

## 3.3 ABLATIONS AND EXTENSIONS

To provide a comprehensive evaluation of NORMBT, we extend our study to include additional training techniques and scaling variants: (1) LayerNorm, (2) global gradient clipping, (3) active-learning style scaling, and (4) ablations of NORMBT and its integration into BT-based variants in Section 3.1. The results are reported in Table 2.

**LayerNorm & Gradient Clipping.** A natural question is whether normalizing the representation can mitigate distance-induced imbalance, rather than normalizing the gradient contributions. We therefore apply a LayerNorm before the final reward layer. While this yields mild gains in some categories, it underperforms on fine-grained tasks. The key limitation is that LayerNorm stabilizes embedding scale but does not modify pairwise representation distances, leaving BT gradient dynamics unchanged. Similarly, global gradient clipping avoids extreme gradients, which improves optimization stability but does not address the relative weighting across pairs.

**Active-learning Scaling.** Prior active learning works (Shen et al., 2025; Feng et al., 2025) prioritize large-distance pairs to improve sample efficiency. Appendix H outlines the key differences between their setting and ours. Nevertheless, for empirical comparison, we adapt the D-opt score formulation (Shen et al., 2025) and extend it with EMA to use as continuous per-pair weights. Although this weighting improves Safety ($77.97 \rightarrow 78.99$), it significantly reduces Reasoning performance ($75.41 \rightarrow 72.90$), consistent with our view that emphasizing large-distance pairs suppresses fine-grained distinctions. In contrast, NORMBT improves Reasoning while maintaining competitive performance in other categories.

**NORMBT Ablations.** We first ablate on similarity measures used for scaling, replacing the last-token embedding L2-distance with (1) cosine similarity and (2) the L2-distance of average-pooled embeddings. Both alternatives underperform compared to NORMBT. This aligns with our theoretical motivation: the last-token embedding is the direct input to the reward head, making its L2 difference an effective proxy for the gradient of the BT score (Eq. 6). Alternative similarity metrics do not reflect this gradient structure. We then remove EMA to show performance degradation as the scale of representation distance drift. Lastly, we apply NORMBT on all BT-based variants and observe consistent improvements across all counterparts.

Table 2: Further studies; RMs are trained from gemma-2b-it on Unified-Feedback (80K).

| Reward model | Average | Chat | Chat-Hard | Safety | Reasoning |
|---|---|---|---|---|---|
| BT (baseline) | 72.25 | 95.25 | 40.35 | 77.97 | 75.41 |
| BT (LayerNorm) | 72.25 | 94.97 | 41.34 | 79.46 | 73.22 |
| BT (Grad. clip=3.0) | 72.25 | 94.97 | 41.34 | 79.46 | 73.22 |
| BT (Grad. clip=5.0) | 72.69 | 95.53 | 39.69 | 77.57 | 77.96 |
| D-opt Scaling | 72.30 | 96.51 | 40.79 | 78.99 | 72.90 |
| NORMBT (No EMA) | 67.78 | 94.97 | 35.31 | 75.81 | 65.04 |
| NORMBT (Avg. pool) | 69.68 | 96.09 | 38.16 | 73.51 | 70.97 |
| NORMBT (Cos. sim.) | 71.88 | 95.53 | 39.47 | 77.16 | 75.37 |
| NORMBT | 73.57 | 95.81 | 39.80 | 77.97 | 80.71 |
| BT + margin | 71.23 | 95.81 | 37.50 | 78.65 | 72.98 |
| NORMBT + margin | 72.30 | 95.81 | 38.82 | 77.57 | 77.02 |
| BT + margin out | 72.53 | 96.09 | 38.38 | 77.57 | 78.09 |
| NORMBT + margin out | 73.36 | 96.65 | 38.82 | 77.16 | 80.83 |
| BT + label smooth | 69.95 | 93.85 | 37.72 | 75.95 | 72.28 |
| NORMBT + label smooth | 73.46 | 95.53 | 39.25 | 78.11 | 80.96 |

## 3.4 DOWNSTREAM RLHF

To evaluate the practical utility of reward models, we assess their performance on Best-of-N (BoN) response selection. Unlike standard benchmarks that measure binary accuracy, BoN directly reflects how an RM guides a policy model among diverse candidate responses, closely mirroring usage in downstream RLHF pipelines.

Following prior works (Gao et al., 2022; Coste et al., 2023; Yang et al., 2024), we conduct BoN sampling on a 1K held-out test set. For each prompt, we sample $n$ candidate responses, score them using the trained RM and select the highest-scoring response. We then use a gold reward model to evaluate the selected responses, and report the gold score averaged over the 1K prompts. This reflects the true quality of the responses selected by RM. We set the number of responses $n$ ranging from 1 to 402 for each prompt. This roughly responds to the KL-divergence of 0 to 5 from policy model, according to the equation $\text{KL}_{\text{BoN}} = \ln n - \frac{n-1}{n}$. The gold score model[3] is a 7B RM finetuned on the entire Unified-Feedback dataset, following Yang et al. (2024).

The results are reported in Figure 5. NORMBT consistently achieves higher gold scores, outperforming all BT baselines. Moreover, applying NORMBT on top of each BT variant yields further improvements, most notably for label smoothing. This demonstrates that NORMBT complements and strengthens existing approaches in ways that carry over to downstream RLHF applications.

---

[3]reward-model-Mistral-7B-instruct-Unified-Feedback

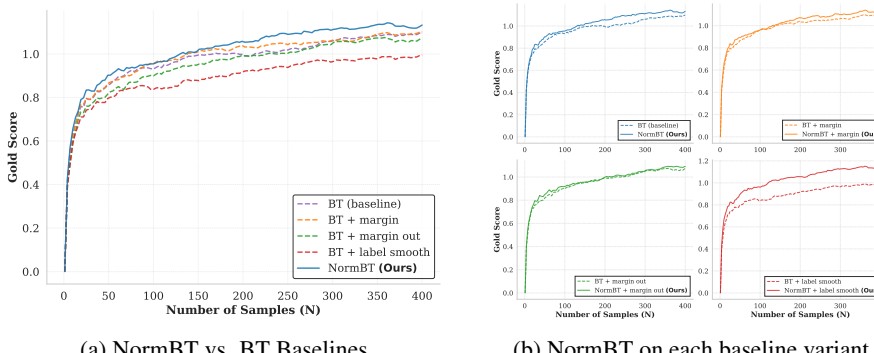

(a) NormBT vs. BT Baselines      (b) NormBT on each baseline variant

Figure 5: **Best-of-N Selection.** Higher gold score indicates higher quality of the selected responses. NORMBT consistently outperforms all BT baselines, as well as BT counterparts in ablation studies.

## 4 RELATED WORKS

**Reward Model Paradigms.** Reward modeling is central to reinforcement learning from human feedback (RLHF), aligning LLMs with human preferences (Christiano et al., 2017; Stiennon et al., 2022; Bai et al., 2022; Dong et al., 2024). Reward models can be broadly classified into paradigms. The first is discriminative RM, typically with a linear score head trained via a Bradley-Terry (BT) style objective (Wang et al., 2025; Liu et al., 2024a; Yang et al., 2024). Another line of research focuses on generative RMs, aiming to leverage the model's generation ability to produce rationales, critiques, or verifier signals (Mahan et al., 2024; Zhang et al., 2025; Chen et al., 2025b; Zhu et al., 2025; Chen et al., 2025a). Beyond this dichotomy, several branches enrich the supervision signal, such as providing fine-grained feedback (Wu et al., 2023), incorporating multi-objectives (Wang et al., 2024), and providing awareness of distribution or uncertainty (Dorka, 2024; Sun et al., 2025b). In this work, we focus on reward models in the discriminative, BT-based paradigm. While Direct alignment methods (Rafailov et al., 2024; Meng et al., 2024; Gupta et al., 2025) bypass explicit reward modeling, they implicitly rely on similar BT-style pairwise signals. Our work provides a principled analysis of how pairwise signals translate into parameter updates.

**Improvements on Reward Modeling.** Despite the success of reward models in RLHF, a broad literature targets complementary weaknesses in training and deployment. For instance, Liu et al. (2024b) incorporates ties into BT models, while Coste et al. (2023) identifies the issue of overoptimization. Yang et al. (2024) shows that regularizing hidden states helps generalization against distribution shifts. Data-centric improvements curate higher-quality comparisons to enhance reward modeling (Liu et al., 2024c; Cui et al., 2023; Liu et al., 2025). A parallel line of work improves evaluation protocols, aiming to better capture the practical utility of reward models (Liu et al., 2024d; Frick et al., 2024; Malik et al., 2025). Our study is orthogonal to these directions, where we address a structural limitation in BT-style updates without modifying data or model architecture. Hong et al. (2025) also analyzes representation distance in RMs and introduces zero-centered reward regularization. Our work analyzes how each pair directly contributes to model updates and proposes to modify the BT gradients explicitly. Finally, theoretical and diagnostic studies (Razin et al., 2025; Sun et al., 2025a) analyze fundamental drawbacks of reward models. Our study is complementary to these works in exploring the limitations as well as further possibilities of the BT objective.

## 5 CONCLUSION

In this study, we analyze the widely used Bradley-Terry loss for reward modeling and identify a key limitation in its update dynamics. We show that update magnitude depends jointly on (1) prediction error, and (2) representation distance between the response pair, with the latter introducing biased learning signals that are misaligned with model performance. To this end, we propose NORMBT, as a lightweight modification to BT-loss through pair-wise normalization. Experiments across various base models and training datasets show consistent performance gains of NORMBT over the standard BT objective. Our findings provide insights into extracting faithful and efficient learning signals, thereby facilitating preference modeling and LM alignment.

## 6 ETHICS STATEMENT

Our work on reward modeling is committed to responsible AI development and adheres to standard academic and ethical practices. This particular project does not involve human subjects or raise concerns regarding data privacy, bias, or fairness in its current scope. Our research focuses on foundational architectural and training methodologies, with no direct application to the creation of sensitive or harmful queries. We are dedicated to ensuring that our research contributes to the safe and beneficial advancement of AI and are actively exploring methods to detect and prevent malicious applications of LLM alignment.

## 7 REPRODUCIBILITY STATEMENT

To ensure reproducibility of our results, we provide the following resources: (1) complete implementation details and hyperparameters are described in Sec 3 and Appendix B; (2) all benchmarks and models used in our experiments are publicly available and properly cited with access information provided in Sec 3; and (3) source code will be made available upon publication to facilitate replication of our experimental results.

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

## A  LLM USAGE

We used a large language model (LLM) for assistance, primarily for polishing paper writing. Following its generation, the authors carefully reviewed, edited, and rewrote the content to ensure its accuracy and alignment with the paper's standards.

## B  IMPLEMENTATION DETAILS

**Baseline Details.**  This section includes the formulation for each baseline in Sec 3, as well as their relevance to our method.

- *BT Baseline.* The standard Bradley–Terry objective in Eq 1.
- *BT + Margin.* (Touvron et al., 2023) For the Unified-Feedback dataset, which provides ratings for the responses, we extract a ground-truth margin $m$ (i.e., difference in ratings) and incorporate it into the BT loss. In this formulation, the margin is introduced inside the log-sigmoid:

$$\mathcal{L}_{\text{Margin}}(\theta) = -\mathbb{E}_{(x, y_w, y_l, m) \sim D} \left[ \log \sigma \left( r_w - r_l - m \right) \right]. \tag{14}$$

  This variant shifts the effective decision boundary for each pair of chosen and rejected responses according to the ground-truth margin. This baseline tests the effect of injecting ground-truth-based error strength alone, without modifying BT's representation coupling.

- *BT + Margin (outside).* (Wang et al., 2025) In the second formulation, the margin is applied outside of log-sigmoid as a multiplicative weight on the per-sample loss:

$$\mathcal{L}_{\text{Margin(out)}}(\theta) = -\mathbb{E}_{(x, y_w, y_l, m) \sim D} \left[ m \cdot \log \sigma \left( r_w - r_l \right) \right]. \tag{15}$$

  This objective emphasizes the role of prediction error by upweighting pairs with larger ground-truth margins. It tests the effect of a reweighting scheme driven by ground-truth labels, rather than representation distance.

- *BT + Label Smoothing.* (Liu et al., 2024a) A widely used regularization technique in classification models, where soft labels replace hard binary targets:

$$\mathcal{L}_{\text{LS}}(\theta) = -\mathbb{E}_{(x, y_w, y_l) \sim D} \left[ (1 - \alpha) \log \sigma \left( r_w - r_l \right) + \alpha \left( r_l - r_w \right) \right]. \tag{16}$$

  Differentiating w.r.t. parameters, its gradient is given by

$$\nabla_\theta \mathcal{L}_{\text{LS}} = \left( \sigma(d) - (1 - \alpha) \right) \cdot \left( \nabla_\theta r_w - \nabla_\theta r_l \right), \tag{17}$$

  where $\alpha = 0$ recovers the BT objective. Note that since $\sigma(d) \in [0, 1]$, its norm is

$$\begin{aligned} \left\| \nabla_\theta \mathcal{L} \right\| &= \left| \sigma(d) - (1 - \alpha) \right| \left\| \nabla_\theta r_w - \nabla_\theta r_l \right\| \\ &= \left( \underbrace{1 - \sigma(d)}_{\text{BT magnitude}} - \alpha \right) \left\| \nabla_\theta r_w - \nabla_\theta r_l \right\|. \end{aligned} \tag{18}$$

  From this perspective, label smoothing adjusts the strength of BT gradient from the prediction error term, by further reducing its magnitude by $\alpha$. This offers another relevant and generic method for our comparison.

In summary, the margin-based variants modify the prediction-error term by incorporating ground-truth reward differences, effectively adjusting update strength using external supervision. Label smoothing also acts on the prediction-error term, but does so by uniformly reducing its magnitude across all pairs. These approaches serve as natural baselines that manipulate error strength, while our method addresses a distinct factor by normalizing representation distance.

**Training Details.**  We implement all methods based on transformers (Wolf et al., 2020) and trl (von Werra et al., 2020), developing from the work of GRM (Yang et al., 2024). To use the Unified-Feedback dataset, we downsample the training data from the "all" set. All reward models are trained with the default reward head, as a linear layer with shape (hidden size, 1). For NORMBT, the normalization strength $\alpha$ is set to 1 by default. For label smoothing, $\alpha$ is set to 0.1 by default. For

all BT baselines, we conduct an extensive grid search over learning rates and report the best-tuned results. For BT-based variants (margin, margin-out, label smoothing), we search over a range of learning rates centered around the best value found for BT and report the best configuration. All models are trained for 1 epoch with full parameter tuning. We truncate the inputs over 4096 tokens, and an effective batch size of 256 with gradient accumulation. Our experiments are conducted using NVIDIA RTX A6000 49G.

Table 3: Key hyperparameter details in reward model training.

| Basic information | |
|---|---|
| Datasets | Unified-Feedback (80K), Skywork-Reward-Preference-80K-v0.2 |
| Base models | gemma-2b-it, Llama-3.2-3b-Instruct |
| Quantization for training | bf16 |
| Optimizer | sgd |
| Momentum | 0.9 |
| Batch size | 256 |
| Gradient accumulation | 4 |
| Learning Rate Scheduler | cosine |
| Warmup Ratio | 0.03 |

## C  EXPERIMENT ERROR ANALYSIS

To assess the robustness of our main results reported in Table 1, we conduct an expanded evaluation on five random seeds for each method. Using a 40K subset of the Unified-Feedback dataset, we train both NORMBT and the BT baseline following the same training configuration described in Appendix B. We compare these 10 models to illustrate the performance improvement of NORMBT over the BT baseline. The results reported in Table 4 and Figure 6 are consistent with the main experiments, showing higher average scores across the majority of RewardBench and particularly strong gains in the Reasoning category.

| Reward Model | Average | Chat | Chat Hard | Safety | Reasoning |
|---|---|---|---|---|---|
| BT (Baseline) | 68.15 | 94.69 | 37.48 | 75.11 | 65.34 |
| NORMBT (Ours) | 69.62 | 95.03 | 38.00 | 74.95 | 70.50 |

Table 4: **Comparison of Ten Models.** Mean performance on RewardBench over models trained on five random seeds aligns with the main results reported in Table 1.

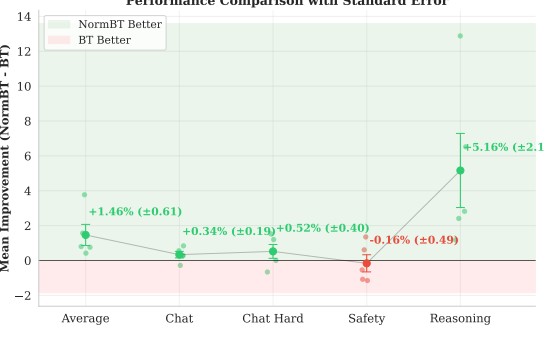

(a) RewardBench

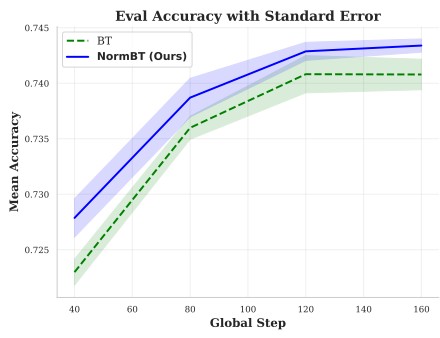

(b) Unified-Feedback Eval Accuracy

Figure 6: **Comparison on Ten Models.** Detailed performance of models trained on five random seeds with standard errors on (a) RewardBench categories, and (b) Unified-Feedback Evaluation.

# D  ADDITIONAL EXAMPLES

**Small Representation Distance Pairs.**    This section follows up on the performance comparison between BT Baseline and NORMBT in Figure 4. We provide example instances of "Small-Distance" response pairs from RewardBench where NORMBT predicts correctly while BT Baseline does not. This gives a clear illustration to pinpoint the preference pairs where NORMBT gives the largest performance gains.

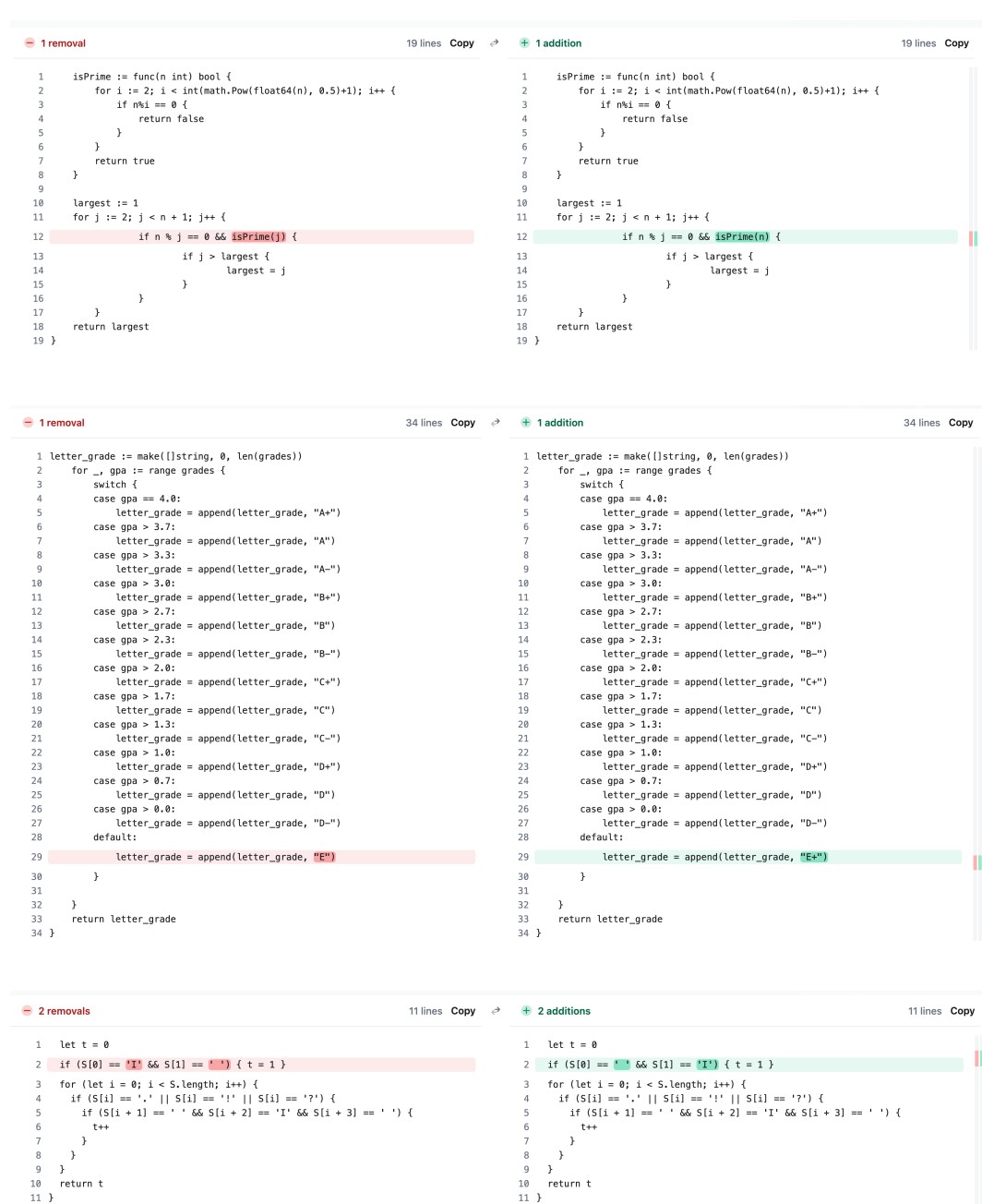

Figure 7: **Small-Distance Pairs.** Example response pairs from RewardBench, where NORMBT predicts correctly on while BT Baseline does not. These pairs fall within the "Small-Distance" range. Response pairs from three different prompts are shown, where left shows the chosen the right shows the rejected response.

# E  FULL VS. PROXY REPRESENTATION DISTANCE

This section analyzes empirically the relationship between the full representation distance $\left\lVert \nabla_\theta(r_w - r_l) \right\rVert$ and its proxy $\lVert h_w - h_l \rVert$ as defined in Sec 2.1. Figure 8 plots the correlation between these two quantities. For gemma-2b-it, the correlation is strong with $r = 0.928$, suggesting that the embedding difference is a highly reliable stand-in for the full representation distance. For Llama-3.2-3b-Instruct, the correlation is more modest at the beginning of reward model training ($r = 0.682$; shown in Figure 8b) but increases to $r = 0.932$ after some initial training. This may indicate that the proxy becomes more accurate once the pretrained model begins adapting to the reward modeling objective.

These observations support the theoretical analysis in Eq. 7, which establishes the connection between $\left\lVert \nabla_\theta(r_w - r_l) \right\rVert$ and $\lVert h_w - h_l \rVert$. This suggests the strategy to utilize $\lVert h_w - h_l \rVert$ as a proxy is a computationally efficient and empirically validated substitute for the full distance, avoiding costly backward passes.

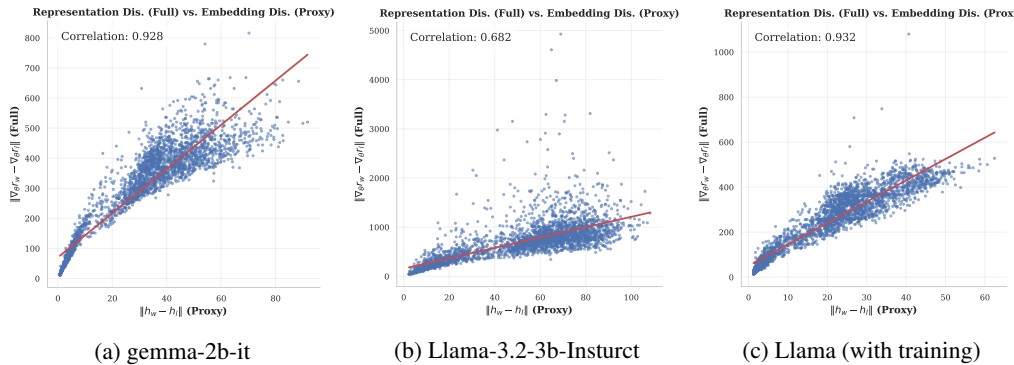

| (a) gemma-2b-it | (b) Llama-3.2-3b-Insturct | (c) Llama (with training) |

Figure 8: **Correlation of Full vs. Proxy Representation Distance.** Both base models show a strong correlation ($r > 0.9$) between the two quantities.

# F  REPRESENTATION SCALE & EMA

One challenge in NORMBT normalization arises from the non-stationary behavior of representation distances $\lVert h_w - h_l \rVert$ during training of the reward model. Since the LLM backbone is jointly optimized with the reward head, the learned representation space could evolve considerably.

Figure 9 illustrates this observation: the mean representation distance shifts significantly over the course of optimization, while the variance remains consistently large. And the same behavior is observed across both Unified-Feedback and Skywork-Reward-Preference-80K-v0.2 datasets, as well as for all BT baselines. As a result, although the large variance highlights the potential gradient imbalance, any normalization scheme that relies directly on raw representation magnitudes inherits the instability created by scale shifts.

To mitigate this issue, we incorporate an Exponential Moving Average (EMA) as stated in Eq 12. By tracking a running mean of representation distance, EMA provides a stable and adaptive reference point that smooths out fluctuations in embedding scale while remaining responsive to long-term trends. By normalizing each pair relative to this moving reference, rather than to its raw magnitude, we ensure that weighting reflects relative distinctions among pairs. This yields consistent gradient scaling across training. The benefit of EMA is also reflected in the performance gap observed in ablation studies shown in Table 2.

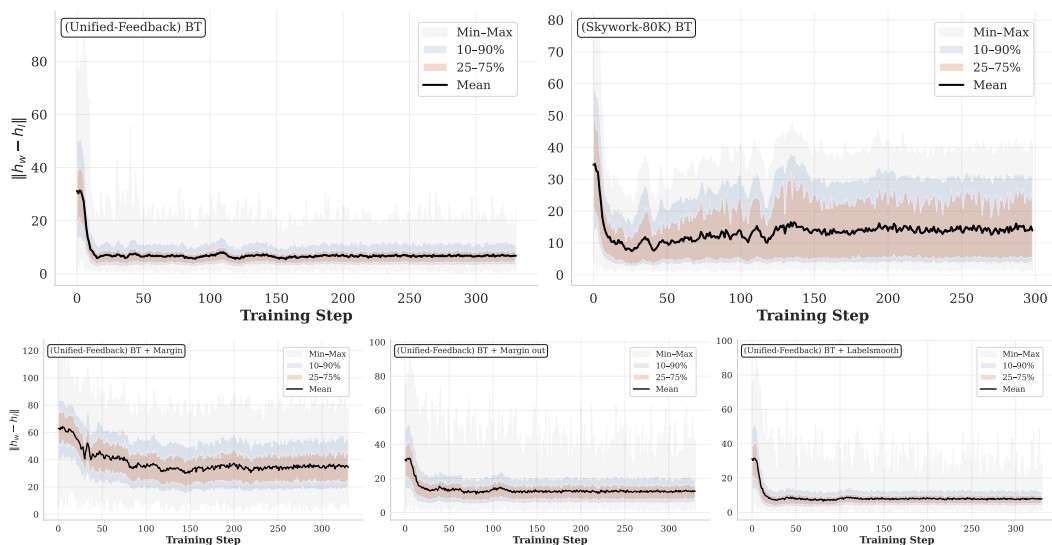

Figure 9: **Dynamics of Representation Distance** during RM Training for standard BT on Unified-Feedback and Skywork-80K (top), and all BT variants on Unified-Feedback (bottom). For each method, we report the mean, 25–75 percentile, 10–90 percentile, and full min–max range. Across all settings, the mean distance shifts substantially, while the variance remains high.

# G  LIMITATIONS & FUTURE WORK

**Method.** While NORMBT yields consistent gains on fine-grained distinctions, a slight trade-off remains in larger-distance regions; more intricate scaling strategies may help mitigate this. Unified-Feedback and Skywork-80K are human-annotated or curated datasets, and settings with substantial noise or near-duplicate pairs may require additional handling.

**Evaluation.** Our comparison focuses on BT-based variants. Techniques involving regularization or auxiliary objectives may alter gradient dynamics beyond Eq. 7, and exploring how NORMBT interacts with such methods is an interesting direction. We use Best-of-N to mirror downstream RLHF usage; full policy-level evaluations (Schulman et al., 2017; Ahmadian et al., 2024) would offer a more comprehensive assessment.

**Analysis.** Our analysis centers on gradient magnitudes to quantify each pair's contribution to model updates. Incorporating gradient-direction or correlation analysis would provide complementary insight into the underlying representation space.

# H    INTUITION: LARGE- VS. SMALL-DISTANCE PAIRS

Small representation distance can arise from two reasons: (1) **uninformative or noisy pairs**, such as near-duplicate responses that convey little preference signal; or (2) **difficult but meaningful pairs**, where the backbone fails to separate two responses despite clear human preference signal (e.g., correct vs. subtly flawed code). As illustrated in Figure 2, such case (2) pairs are common in practice, and Figure 7 provides concrete examples.

This distinction matters because BT loss treats both cases identically: small-distance pairs always receive small gradients. While suppressing case (1) is desirable, suppressing case (2) is harmful, as these are precisely the pairs containing the fine-grained distinctions that the reward model should learn but inherently struggles to separate with the current $\Phi$.

In our datasets (UnifiedFeedback and Skywork-80K), most pairs are human-annotated or carefully curated, and following Yang et al. (2024), pairs with identical ratings are filtered out. This substantially reduces the prevalence of case (1), making small-distance pairs far more likely to reflect case (2) of the difficult, fine-grained distinctions that RM should learn. Viewed through this lens, NORMBT effectively *upweights hard, meaningful pairs* whose gradients would otherwise be suppressed due to representation limitations in the current model.

# I    RELATION TO ACTIVE LEARNING APPROACHES

The above intuition also clarifies how our analysis relates to active learning approaches such as PILAF (Feng et al., 2025) and ARM (Shen et al., 2025), which prioritize **large-distance pairs** to improve sample efficiency. While this appears at odds with our motivation, the two settings address fundamentally different problems.

Active-learning methods focus on **sample selection**: given a noisy pool of model-generated responses, identify pairs that are maximally informative. In this setting, many candidates reflect case (1) from Appendix H as uninformative or trivial pairs that provide little preference signal. Moreover, ARM adopts an embedding-based RM with a *frozen* LLM backbone, meaning $\Phi$ is assumed reliable and representation learning is not part of training. Under these assumptions, difficult but meaningful small-distance pairs cannot be corrected, thus selecting large-distance pairs becomes a principled strategy for sample efficiency.

In contrast, sequence-classifier RMs jointly train the backbone and reward head. The assumption of fixed, reliable $\Phi$ no longer holds, especially as the model newly adapts to the task. Therefore, a small-distance pair could arise because the backbone has not yet learned to separate meaningful distinctions (i.e., case (2)), rather than a lack of meaningful preference information. These are precisely the pairs where the reward model needs stronger updates. Yet under standard BT loss, *both case (1) and case (2) receive small gradients, creating a learning bottleneck.*

This distinction is amplified in our datasets. UnifiedFeedback and Skywork-80K contain human-annotated or carefully curated pairs, and identical-rating pairs are filtered out following Yang et al. (2024). This substantially removes case (1) examples, making small-distance pairs far more likely to represent meaningful but difficult decisions. In this regime, prioritizing large-distance pairs (as active-learning methods do) suppresses the critical fine-grained distinctions.

Our empirical results also reflect this: active-learning style D-opt scaling underperforms in our setting (Table 2), since the difficult, fine-grained distinctions driving performance in Reasoning are further suppressed. In contrast, NORMBT improves these regions while maintaining competitive performance elsewhere. However, we also note that our D-opt scaling uses a heuristic continuous relaxation for fair comparison using the same dataset and ordering. This differs from its original use as a sample-selection criterion, and we include it only to highlight the conceptual differences between active-learning approaches and NORMBT in the context of reward model optimization.

