# OpenReview forum: "When Distance Distracts: Representation Distance Bias in BT-Loss for Reward Models"
_ICLR.cc/2026/Conference — Submitted to ICLR 2026_

### Official Review · Reviewer_31NW · 2025-10-19

**Soundness:** 3
**Presentation:** 3
**Contribution:** 2
**Rating:** 2
**Confidence:** 4

**Summary:**

The authors observed that the gradient update under BT loss involving representation distance and proposed an alternative loss that normalize over representation distance so that mean training signal is from prediction accuracy.

**Strengths:**

The presentation is good and derivations sound. Empirical results showed improvement over existing baselines in at least some of the experiments.

**Weaknesses:**

The paper builds on the premise that representation distance ***should not*** be treated as part of the signal and its apparance in gradient is a problem that we should solve. This key premise is not justified to a satisfactory level. In fact from statistical literature on logistic regressions (where a linear BT is a special case), the asymptotic variance of parameter should depends on representation distance (e.g., through [classic asymptotic theory](https://stats.stackexchange.com/questions/231329/fisher-information-in-logit-model)). This is the key that the model can do prediction on unseen pairs. I acknowledge the empirical results showed improvements in some cases but it is not consistent in all experiments and the conceptual connection is unclear. If the premise is true, following the authors' argument, for any classification problem use cross entropy loss the last layer representation vector (in place of difference of representation) should not appear in the gradient as BT can be seen as a classification problem with features being representation *difference* before last layer.

Also the experimental results in tables is hard to interpret by themselves. I noticed that in the tables the authors highlight the proposed method without explaining, while the proposed method is neither highest nor lowest in the numerical results, e.g., in Table 1, with gemma-2b-it, in task **Chat** BT + margin out is largest and BT+ label smooth is the smallest. Same thing happens in every individual tasks beside **Reasoning** (and average due to Reasoning).

**Questions:**

- Intuition on representation distance and prediction error entanglement:
    - The authors argued in Q1 that it supports intuition from (7) and argued that representation distance is entangled with prediction error by the product structure in the gradient. However, it can be deeper than that product structure. If we are willing to assume the reward model is smooth enough (as the authors assumed), pairs close in representation space should also have similar reward value and small $|d|$. In my understanding these are pairs that are *not* informative for the model: in the extreme case that the two responses are identical so $d=0$, preference annotation is at best a coin toss, the labeling error $\sigma(d)-1$, in such data, is going to be large either way, and the reward distance **should** kill that data in update as preference between identical pairs are pure noise. This argument should holds for small distance pairs --- model (and us) should not be surprised that we cannot predict results of close-to-fair coin flips well as there is no signal. In this case should the representation distance be used to reduce the weight on these unsurprising pairs? The authors proposal would up weight these kind of high noise pairs.

    - Aren't representation part of the signal? For reward modeling, getting the ranking of existing pairs is not the end of the story, one needs to be able to predict unseen new pairs. These can be done leveraging smoothness in representation space (Sun et al. 2025). For reward model to generalize to unseen pairs we would need to see things large enough to cover most of the representations (difference) space so that we are doing interpolation rather than extrapolation. What is the intuition that we should kill the signal from representation space?

- What is the mechanism the proposed normalization on **loss** works?
    - It is indeed true that for a data point that 1) the model gets wrong and 2) the representation distance is large. The loss gradient can even diverge. Is the normalization avoiding this only? Will a gradient clipping achieve the same goal? What is the percentage of data point these two things happen at the same time?
        - This is close to ask that whether the issue of having distance in gradient is 1) it just should not be there or 2) some of them can be too big that harm optimization.
    - A layer norm before last linear layer. How does the authors' proposal compare to this implementation tweak? From the statistical literature it is know that it can be beneficial to normalize features before fitting logistic regression.

- How to reconcile with past active learning results?
    - A line of work on active learning like Feng et al. 2025 and Shen et al. 2025 actually treated representation distance as part of the signal and look for points that the model struggles and representation is far. They showed doing this is beneficial. My intuition is that these design helped to cover large representation space for generalization. I wonder how these line of results can be reconciled with what the authors observed that normalizing using representation distance being helpful?




Sun, Hao, Yunyi Shen, and Jean-Francois Ton. "Rethinking reward modeling in preference-based large language model alignment." In The Thirteenth International Conference on Learning Representations. 2025.

Feng, Y., Kwiatkowski, A., Zheng, K., Kempe, J. and Duan, Y., 2025. Pilaf: Optimal human preference sampling for reward modeling. arXiv preprint arXiv:2502.04270.

Shen, Y., Sun, H. and Ton, J.F., Active Reward Modeling: Adaptive Preference Labeling for Large Language Model Alignment. In Forty-second International Conference on Machine Learning. 2025

---

> ### Author Response · Authors · 2025-11-30
> **Response to Reviewer 31NW (1/3)**
>
> Thank you for the thoughtful and insightful feedback. We have revised our paper based on your comments. We detail our response below to address your concerns.
>
> >**W1**: If the premise is true, for any classification problem use cross entropy loss the last layer representation vector (in place of difference of representation) should not appear in the gradient as BT can be seen as a classification problem with features being representation difference before last layer.
>
> **A1**: We believe the **pairwise difference** in representations is exactly what makes reward modeling with BT fundamentally distinct from standard classification. In standard cross-entropy classification, the gradient for an example depends only on its own representation, which reflects properties of that input alone. In contrast, BT cross-entropy loss couples each update to **two** embeddings through the feature vector $h_w-h_l$, so the update magnitude depends on their relative position in representation space. This creates a structural dependence not present in standard CE classification.
> As shown in Eq. (7), the BT update size is jointly determined by prediction error and the embedding distance between the two responses of that pair. A direct consequence is that two mis-ranked pairs with identical prediction error can receive very different gradient magnitudes solely due to differences in $\|| h_w-h_l \||$. We elaborate in Q1 why this coupling is not ideal for learning. But crucially, this behavior does not arise in standard CE classification, where the gradient for an example is not affected by the representation of another example. Therefore, the gradient bias we analyze originates specifically from the **pairwise BT formulation**, rather than a general property of cross-entropy losses.
>
> >**W2**: The experimental results in tables is hard to interpret by themselves. Authors highlight the proposed method without explaining, while the proposed method is neither highest nor lowest in the numerical results. In Table 1, with gemma-2b-it, in task Chat BT + margin out is largest and BT+ label smooth is the smallest. Same thing happens in every individual tasks beside Reasoning (and average due to Reasoning).
>
> **A2**: We will clarify in the caption that the highlighted row distinguishes our method from BT baselines. However, we would like to point out that BT-based variants show significant trade-offs. For example, BT + margin improves Safety (77.97 → 78.65) but hurts both Chat-Hard (40.35 → 37.50) and Reasoning (75.41 → 72.98). In contrast, NormBT yields consistent gains in Reasoning (75.41 → 80.71, the largest improvement) while remaining comparable on other categories.
>
> We attribute this to improved BT signals for fine-grained distinctions: as shown in Figure 2, the Reasoning category exhibits the smallest distance. Under BT, the small-distance region receives inherently weak updates, as shown in Eq. (7).  NormBT explicitly alleviates this by upweighting learning signals for small-distance regions. As a result, the largest gains naturally appear in categories dominated by fine-grained distinctions, such as Reasoning. We will also note in Limitations that a slight trade-off remains and discuss exploring more sophisticated scaling strategies.

---

> ### Author Response · Authors · 2025-11-30
> **Response to Reviewer 31NW (2/3)**
>
> >**Q1**: Small distance pairs are not informative, and reward distance should kill that data in update as there is no signal
>
> **A1**: Thanks for your thoughtful feedback. We believe this intuition holds when the feature map $\Phi$ is well-trained. However, whether large-distance pairs are "more informative'' depends critically on the feature map $\Phi$  reliably distinguishing good vs. bad samples. In general, a small representation distance can arise from two reasons: **(1) uninformative or noisy pairs**, e.g., identical or near-identical responses as mentioned, and **(2) difficult pair**, where the current backbone fails to separate two responses that have clear preference signals, e.g., correct vs. flawed code. Figure 2 shows such case (2) pairs are common, and Figure 7 provides concrete examples demonstrating their relevance.
>
> Thus in training of RM with joint optimization of $\Phi$ and reward head, a small embedding distance could also reflect a limitation of the backbone, not just lack of meaningful preference information. However, both cases receive small updates under BT loss. While downweighting case (1) is appropriate, case (2) is precisely where the model needs strong learning signals.
>
> In our datasets (UnifiedFeedback and Skywork-80K), most pairs are human-annotated or carefully curated, and following Yang et al.[1], pairs with identical ratings are filtered out. This substantially reduces case (1), making small-distance pairs far more likely to reflect case (2) of the difficult, fine-grained distinctions that RM should learn. From this perspective, NormBT can be interpreted as upweighting difficult pairs whose gradients are otherwise inherently weak due to limitations in current model's representation. We will add this clarification, as well as note in Limitations that datasets containing many noisy or near-duplicate pairs may require additional handling.
>
>
> >**Q3**: For reward model to generalize to unseen pairs we would need to see things large enough to cover most of the representations (difference) space so that we are doing interpolation rather than extrapolation. What is the intuition that we should kill the signal from representation space?
>
> **A3**: This is a valid concern that involves two different aspects: (1) coverage of representation-difference space provided by dataset, and (2) *effective* coverage by optimization under BT.
>
> Given a fixed training dataset (in our setting UF or Skywork), coverage arises from the distribution of training data. However, the effective coverage (i.e., *which regions of this space the model actually learns from*) is driven by gradient magnitudes during optimization. When BT gradients scale with representation distance, optimization is systematically biased toward already-separated regions of the distribution. This reduces effective coverage over small-representation or difficult regions. By normalizing, we ensure all regions of representation space receive learning signals proportional to prediction error. We believe that in this sense, our method utilizes representation signals to address the imbalance and enhance effective coverage, rather than discarding the signal altogether.
>
>
> >**Q4**: What is the mechanism the proposed normalization on loss works?
>
> **A4**: (1) Our normalization does not merely avoid large diverging gradients, as upweighting pairs with suppressed gradients is an equally critical branch of NormBT. While **gradient clipping** can improve general optimization stability, it does not address the BT structural imbalance and leaves the relative weighting across pairs unchanged. To demonstrate the comparison, we apply gradient clipping with max_grad_norm ∈ {3.0, 5.0}, and report the best-tuned results over a range of LRs in the following table.
>
>  (2) **LayerNorm** controls embedding scale but does not necessarily modify the relative pairwise representation distance. The BT gradient dynamics remain unchanged under LayerNorm, and thus does not substitute for our method. We add LayerNorm before the last linear layer, and report the best-tuned performance below.
>
> | Method         | Average | Chat  | Chat Hard | Safety | Reasoning |
> |----------------|---------|-------|-----------|--------|-----------|
> | Grad. Clip (max=5.0)     | 72.69   | 95.53 | 39.69     | 77.57  | 77.96     |
> | Grad. Clip (max=3.0)     | 72.25   | 94.97 | 41.34     | 79.46  | 73.22     |
> | LayerNorm      | 72.25   | 94.97 | 41.34     | 79.46  | 73.22     |
> | NormBT (Ours)  | 73.57   | 95.81 | 39.80     | 77.97  | 80.71     |
>
> Standard techniques such as gradient clipping and LayerNorm do not match the improvements achieved by NormBT, confirming that our method addresses a distinct mechanism in BT optimization.

---

> > ### Author Response · Authors · 2025-11-30
> > **Response to Reviewer 31NW (3/3)**
> >
> > > **Q5**: How to reconcile with past active learning results, that treated representation distance as part of the signal and look for points that the model struggles and representation is far?
> >
> > **A5**: Active learning such as PILAF and ARM operate in a different setting and objective from ours. These methods address **sample selection & efficiency**: given *a noisy candidate pool* (e.g., raw LLM-generated responses), identify pairs that are maximally efficient. Our work instead analyzes **optimization dynamics** in training: given a *fixed dataset*, how each pair contributes to parameter updates and how to better extract the available learning signal.
> >
> > In sample-selection settings over raw, potentially noisy LLM-generated responses, **many pairs fall into case (1)**, i.e., noisy pairs as discussed in Q1. Moreover, ARM adapts embedding-based RM, where $\Phi$ comes from a **frozen LLM backbone**. Under this assumption, representation learning is not part of training; difficult but meaningful small-distance pairs cannot be corrected, making it reasonable to prioritize large-distance pairs for sample efficiency.
> >
> > In contrast, **sequence-classifier RMs** jointly train the backbone and reward head. The assumption of reliable $\Phi$ may not hold as model newly adapts to the task. A small-distance could arise because the backbone has not yet learned to separate meaningful distinctions (i.e., case (2)), rather than a lack of meaningful preference signal. These are precisely the pairs where the reward model needs stronger updates. Yet under standard BT loss, *both case (1) and case (2) receive small gradients, creating a learning bottleneck.*
> >
> > This distinction is amplified in our **datasets**. UnifiedFeedback and Skywork-80K contain human-annotated or carefully curated pairs, and identical-rating pairs are filtered out. This substantially removes case (1) examples. As a result, small-distance pairs are far more likely to represent meaningful but difficult decisions. In this regime, prioritizing large-distance pairs (as active-learning methods do) suppresses the critical fine-grained distinctions.
> >
> > To provide a concrete comparison, we implement a heuristic adaptation of ARM's D-opt score to our setting. We utilize D-opt formulation in Eq.4 to calculate the score and similarly extend it with EMA to use as continuous per-pair weights. This enables scaling of gradient contributions based on intuition from ARM, and compares two methods on the same dataset size and ordering. In this setting, while the active learning-based weighting improves Safety (77.97 → 78.99), it degrades Reasoning (75.41 → 72.90). This shows that the difficult, fine-grained distinctions driving performance in Reasoning are further suppressed. Nonetheless, we also note that this is an imperfect extension due to differences in setup. We include a detailed discussion in Appendix I.
> >
> > | Method | Average | Chat  | Chat Hard | Safety | Reasoning |
> > |--------|---------|-------|-----------|--------|-----------|
> > | D-opt  | 72.30   | 96.51 | 40.79     | 78.99  | 72.90     |
> > | NormBT (Ours)  | 73.57   | 95.81 | 39.80     | 77.97  | 80.71     |
> >
> > *Reference:*
> >
> > [1] Yang, Rui, et al. "Regularizing hidden states enables learning generalizable reward model for llms." Advances in Neural Information Processing Systems 37 (2024): 62279-62309.

---

### Official Review · Reviewer_HY6G · 2025-10-23

**Soundness:** 3
**Presentation:** 3
**Contribution:** 3
**Rating:** 6
**Confidence:** 3

**Summary:**

This paper analyzes the update dynamics of the Bradley–Terry (BT) loss commonly used in reward modeling for RLHF.
The authors identify that the gradient norm of BT-loss depends not only on the prediction error (i.e., reward difference between chosen and rejected responses) but also on the representation distance between their hidden states (Eq. 7).
This coupling leads to what the paper calls representation distance bias—pairs with large embedding distances receive disproportionately large updates even when correctly ranked, while small-distance pairs (often in reasoning tasks) receive vanishingly small gradients.
To address this, the paper proposes NormBT, a pair-wise normalization scheme that rescales gradient contributions by the inverse of the representation distance (Eq. 10–13).
Empirical results on RewardBench show consistent improvements over the vanilla BT baseline, particularly in the Reasoning category (+5 % absolute accuracy in Table 1), without major regressions in other categories.

**Strengths:**

1. Clear identification of a structural bias in BT-loss: The decomposition of gradient norm (Eq. 7) elegantly shows that update magnitude scales with both prediction error and representation distance. This theoretical insight provides a solid foundation for understanding how BT-based reward models may fail to learn from fine-grained preference pairs, especially in reasoning-oriented data.
2. Simple, lightweight correction: NormBT is a “drop-in” modification requiring no architectural change.
By reweighting each pair’s contribution with $w_i = 1 / \|h_w - h_l\|$, it effectively balances updates between small- and large-distance pairs (Sec. 2.2). The EMA-based normalization (Eq. 11–12) further ensures numerical stability without noticeable computational cost.
3. Empirical consistency without global degradation: Across four experiment settings (Table 1a–1b), NormBT consistently improves performance, with the largest gain in Reasoning pairs—those identified as suffering most from gradient underflow in Figure 2.
The model does not experience major losses in other domains, suggesting a targeted improvement rather than a crude reweighting.
4. Strong conceptual clarity and visualization:  Figures 1–3 compellingly illustrate the problem: reasoning pairs lie close in representation space and thus yield weak gradients under BT-loss.
The analysis connects qualitative intuition with quantitative evidence.

**Weaknesses:**

1. Limited generality of theoretical analysis: The derivation in Eq. 7 assumes a linear score head $r(x, y) = w_s^T h_\phi(x, y)$ and a Lipschitz-smooth embedding map.
This simplification ignores the nonlinear components of modern reward models (layer normalization, activation scaling, or residual mixing).
The paper does not test whether the same coupling holds under non-linear heads or multi-layer scoring networks.
Therefore, the claimed “representation distance bias” may be an artifact of this specific linear assumption rather than a universal property of BT-loss.
2. Gradient magnitude only—no directional analysis: The study focuses purely on gradient norms (Figure 2, Eq. 7) without considering gradient directions or correlations across pairs.
Representation similarity might also induce gradient alignment bias (e.g., correlated update directions causing slow convergence), but this aspect is not examined.
This omission limits the understanding of whether the issue is truly about update strength or about geometric interference in optimization.
3. Narrow experimental scope:  The evaluation relies solely on RewardBench and two backbones (Gemma-2B-it and Llama-3.2-3B).
While Table 1 shows improvement on Reasoning tasks, Safety and Chat-Hard categories exhibit negligible or even slightly negative changes (e.g., Chat-Hard: 40.35 → 39.80).
No standard deviation or significance testing is reported, making the claimed “5 % improvement” potentially within noise range.
4. Lack of downstream validation in RLHF: The paper focuses exclusively on reward model accuracy without verifying whether these improvements translate to better policy alignment. Unlike On the Robustness of Reward Models for LM Alignment (Hong et al., 2025), which analyzes propagation of reward robustness to RLHF training (Figure 5–7 in that paper), NormBT stops short of such experiments.
This leaves open whether reduced representation bias actually yields more stable or less verbose RLHF outcomes.
5. Scalability and numerical stability concerns: While the paper claims “negligible overhead,” pair-wise distance computation and EMA tracking may introduce instability for large batches or larger backbones (7B–13B). The authors provide no training dynamics such as gradient norm evolution or variance plots to demonstrate convergence stability under normalization. Table 2 shows performance drops when EMA is removed (67.78 avg vs 73.57 with EMA), but does not clarify whether EMA stabilizes or merely rescales.
6. Insufficient ablation and alternative metrics:  The proxy $\|h_w - h_l\|$ is justified through correlation with full gradient distance (r = 0.928 in Appendix C), but no comparison with alternative similarity measures (cosine distance, Mahalanobis, etc.) is provided.
Consequently, it is unclear whether “norm distance” is the optimal normalization factor or merely one convenient proxy.
7. Missing analysis of dataset-dependent behavior: Figure 2 shows that reasoning pairs have smaller distances, but the paper never explains why. Are these due to shorter responses, higher lexical overlap, or semantic similarity? Without analyzing dataset-level statistics, the conclusion risks attributing data-specific phenomena to universal model bias.

**Questions:**

1. Eq. 7 assumes a linear reward head. Would the same distance coupling persist if r(x, y) were computed through a multi-layer MLP or a mixture-of-experts head?
2. Have the authors tested whether using a NormBT-trained RM leads to better alignment during PPO or RLOO training, similar to how BSR (Hong et al., 2025) demonstrated downstream effects?
If not, could this method unintentionally bias generation length or safety preference?
3. Since normalization depends on $∥h₍w₎−h₍l₎∥$, how does NormBT behave if the backbone representation scale changes due to LayerNorm configuration or mixed-precision training?
4. Are there cases where $1/∥h₍w₎−h₍l₎∥$ causes gradient explosion for extremely similar pairs?
How is this mitigated beyond the small constant ϵ introduced in Eq. 11?
5. Could the authors provide standard deviations or confidence intervals for Table 1 results to confirm that the observed gains are statistically significant?
6. How would NormBT perform on other preference datasets (e.g., UltraFeedback, Skywork-Reward-Preference-80K) or under OOD settings with domain shift?

---

> ### Author Response · Authors · 2025-11-30
> **Response to Reviewer HY6G (1/3)**
>
> Thank you for your valuable feedback! We have revised our paper based on your comments. We detail our response below to address your concerns.
>
> >**W1 / Q1**: The paper does not test whether the same coupling holds under non-linear heads or multi-layer scoring networks. Therefore, the claimed “representation distance bias” may be an artifact of this specific linear assumption rather than a universal property of BT-loss.
>
> **A1**: We would like to clarify that our gradient analysis does *not* rely on assumption of linear reward head. In the case of a nonlinear head, the BT gradient still contains the representation distance term (up to a multiplicative factor from Jacobian of nonlinear network). Therefore, the effect arises from the pairwise structure of BT loss itself, rather than an artifact of linearity. Empirically, we include an ablation on RM trained with MLP head, implemented as a two-layer network (hidden size 1024, ReLu activation) replacing the linear reward head. We observe that NormBT outperforms BT with similar improvement in the Reasoning category specially. We use a linear head in the main results because it is standard, the MLP head does not provide obvious gains over the linear layer, and the linear formulation aligns cleanly with our theoretical proxy.
>
> | Reward Model     | Average | Chat  | Chat Hard | Safety | Reasoning |
> |------------------|---------|-------|-----------|--------|-----------|
> | BT (MLP)               | 72.23   | 95.81 | 38.93     | 81.50 | 72.69     |
> | NormBT (MLP)    | 72.42   | 95.67 | 39.25     | 78.78 | 75.99     |
>
>
> >**W2**: Gradient magnitude only—no directional analysis: The study focuses purely on gradient norms (Figure 2, Eq. 7) without considering gradient directions or correlations across pairs.
>
> **A2**: We agree that analyzing gradient directions and correlations would provide additional insight into representation geometry. Our study currently focuses on gradient magnitudes because the analysis specifically quantifies how much each pair contributes to model updates. Since this issue arises independently of whether gradients across pairs are directionally aligned, studying magnitudes captures the phenomenon that NormBT is designed to address. Nonetheless, we believe this is a valuable complementary angle and will discuss this in Future Work.
>
> >**W3 / Q5**: No standard deviation or significance testing is reported, making the claimed “5 % improvement” potentially within noise range.
>
> **A3**: We compare NormBT and BT baseline on 5 random seeds to train 10 models in total on a 40K subset of Unified-Feedback, and report the standard errors. Detailed results and visualizations are reported in Appendix C. The comparison aligns with Table 1a: NormBT consistently outperforms BT, with largest and most stable gains in the Reasoning category.
>
> | Reward Model     | Average | Chat   | Chat Hard | Safety | Reasoning |
> |------------------|---------|--------|-----------|--------|-----------|
> | BT (Baseline)    | 68.15   | 94.69  | 37.48     | 75.11  | 65.34     |
> | NormBT (Ours)             | 69.62   | 95.03  | 38.00     | 74.95  | 70.50     |
> | Avg. Improvement | +1.46% (±0.61) | +0.34% (±0.19) | +0.52% (±0.40) | −0.16% (±0.49) | +5.16% (±2.13) |
>
>
> >**W4 / Q2**: Lack of downstream validation in RLHF.
>
> **A4**: We expand our experiments to include Best-of-N (BoN) response selection. BoN evaluates how well RM guides a policy model among diverse response candidates, a setting closely mirroring RLHF usage. Using 1K held-out prompts, we generate N=1,...,402 responses for each prompt (corresponding to KL≈ 0-5 from the policy model). And we use RM to select the best out of the N candidates, then compare the gold-score of the selected response. For compact presentation, we report the AUC of gold score vs. N to summarize performance across the 402 responses. Full setup details and complete results are reported in new Section 3.4. NormBT consistently outperforms all baselines, including each corresponding BT counterpart in ablation studies. While BoN performance correlates with how RMs rank rollout responses in RLHF, policy-level evaluation (e.g. PPO/RLOO training as in Hong et al.) would provide a comprehensive perspective on downstream behavior. We consider this end-to-end RLHF as valuable future work.
>
> | **Reward Model**                         | **AUC**      |
> |------------------------------------------|--------------|
> | BT (baseline)                            | 391.635897   |
> | BT + margin                              | 395.994544   |
> | BT + margin out                          | 381.741569   |
> | BT + label smooth                        | 357.204557   |
> | **NormBT (Ours)**                        | **408.471380** |
> | **NormBT + margin (Ours)**               | **407.590582** |
> | **NormBT + margin out (Ours)**           | **387.366909** |
> | **NormBT + label smooth (Ours)**         | **411.087926** |

---

> ### Author Response · Authors · 2025-11-30
> **Response to Reviewer HY6G (2/3)**
>
> >**W5 / Q3**: Table 2 shows performance drops when EMA is removed (67.78 avg vs 73.57 with EMA), but does not clarify whether EMA stabilizes or merely rescales. How does NormBT behave if the backbone representation scale changes due to LayerNorm configuration or mixed-precision training?
>
> **A5**: This concern with representation scale change is precisely what EMA aims to address. During training, we observe that the mean representation distance can shift substantially, while the variance remains large (Figure 9). Incorporating EMA addresses this issue by tracking a running mean of representation distance, such that we can normalize each pair relative to this reference. This ensures adaptive per-samples weighting despite representation scale changes due to LayerNorm, mixed precision, or general drift. Without EMA, these scale shifts lead to unpredictable weighting behavior, which explains the performance drop in ablation study. To address your concern, although LayerNorm or mixed-precision training can change the representation scale, the EMA adapts dynamically and ensures stable per-pair scaling throughout training. We add discussion of representation scale and EMA in Appendix F.
>
>
> >**W6**: Insufficient ablation and alternative metrics: The proxy $\|| h_w-h_l \||$ is justified through correlation with full gradient distance (r = 0.928 in Appendix C), but no comparison with alternative similarity measures (cosine distance, Mahalanobis, etc.) is provided.
>
> **A6**: To explore alternative similarity measures, we added (1) L2-norm of average-pooled embeddings, (2) cosine similarity, and reported the results in Table 2. Our proposed NormBT using last-token L2 distance outperforms both alternatives. This matches our theoretical motivation: the last-token embedding is the direct input to the reward head, making its L2 difference an effective proxy for the gradient of the BT score (Eq. 6). Alternative similarity measures do not reflect this gradient structure and perform worse empirically.
>
> | Method    | Average | Chat  | Chat Hard | Safety | Reasoning |
> |-----------|---------|-------|-----------|--------|-----------|
> | avg. pool | 69.68   | 96.09 | 38.16     | 73.51  | 70.97     |
> | cos. sim. | 71.88   | 95.53 | 39.47     | 77.16  | 75.37     |
> | NormBT (Ours)      | 73.57   | 95.81 | 39.80     | 77.97  | 80.71     |
>
>
> >**W7**: Figure 2 shows that reasoning pairs have smaller distances, but the paper never explains why. Are these due to shorter responses, higher lexical overlap, or semantic similarity? Without analyzing dataset-level statistics, the conclusion risks attributing data-specific phenomena to universal model bias.
>
> **A7**: Reasoning pairs tend to have smaller representation distances because many of them are highly similar at the token or structural level, giving **higher lexical overlap**. For example, the chosen solution is fully correct while the rejected solution contains a tiny logical error, yet their surface forms and syntactic structure are similar. This literal similarity makes it harder for the RM to separate them in representation space, resulting in small distances despite representing clear preferences (correct vs. wrong solution). Such pairs with fine-grained distinctions are prevalent in practice. For instance, numerous example pairs are shown in Figure 7. Furthermore, NormBT also improves performance on Best-of-N and three additional benchmarks, indicating that its gains are not a phenomenon specific to RewardBench.
>
> >**Q4**: Are there cases where $\frac{1}{\||h_w-h_l\||}$ causes gradient explosion for extremely similar pairs? How is this mitigated beyond the small constant ϵ introduced in Eq. 11?
>
> **A4**: In our experiments, scaling by $\frac{1}{\||h_w-h_l\||}$ does not cause training instability relative to BT. Two factors mitigate this risk: (1) Both training datasets used consist of high-quality, carefully curated pairs, reducing the presence of near-duplicate or noisy samples. (2) Following GRM [1], we filter out pairs with identical ratings, which removes degenerate or ambiguous comparisons. Empirically, we do not observe instabilities and the small constant ϵ provides additional numerical safety.

---

> > ### Author Response · Authors · 2025-11-30
> > **Response to Reviewer HY6G (3/3)**
> >
> > >**Q6**: How would NormBT perform on other preference datasets (e.g., UltraFeedback, Skywork-Reward-Preference-80K) or under OOD settings with domain shift?
> >
> > **A6**: To evaluate on OOD settings beyond RewardBench, we compare NormBT and BT variant on 3 additional RM benchmarks: RM-Bench [2], RewardBench-2 [3], and PPE [4]. These benchmarks are more challenging than RewardBench, where PPE in particular exhibits better correlation with downstream outcomes in post-RLHF LLM performance. Results show that our methods perform consistently better among these benchmarks.
> >
> > **RM-Bench:**
> > | Reward Model  | Average | Chat  | Math  | Code  | Safety | Easy  | Normal | Hard  |
> > |----------------------|---------|---------|----------|----------|---------|-----------|---------|------------|
> > | BT                     | 57.87   | 44.53 | 54.04 | 50.19 | 82.72  | 84.59 | 59.53  | 29.49 |
> > | NormBT (Ours) | 58.98   | 46.60 | 55.81 | 50.05 | 83.45  | 83.81 | 60.66  | 32.46 |
> >
> > **RewardBench-2:**
> > | Reward Model     | Average | Factuality | Precise IF | Math  | Safety | Focus | Ties  |
> > |------------------|---------|------------|------------|-------|--------|--------|--------|
> > | BT               | 30.08   | 31.26      | 23.75      | 40.44 | 49.56  | 24.24  | 11.22 |
> > | NormBT (Ours)    | 34.17   | 32.53      | 31.87      | 43.17 | 54.00  | 28.69  | 14.76 |
> >
> > **PPE:**
> > | Reward Model     | Average | MMLU Pro | Math  | GPQA  | IFEval | MBPP-Plus | PPE Pref |
> > |------------------|---------|----------|--------|--------|---------|-----------|----------|
> > | BT               | 50.822  | 55.443   | 43.703 | 44.100 | 54.793  | 56.069    | 57.762   |
> > | NormBT (Ours)    | 51.329  | 57.061   | 44.469 | 44.301 | 54.725  | 56.093    | 58.301   |
> >
> >
> > *References:*
> >
> > [1] Yang, Rui, et al. "Regularizing hidden states enables learning generalizable reward model for llms." Advances in Neural Information Processing Systems 37 (2024): 62279-62309.\
> > [2] Liu, Yantao, et al. "Rm-bench: Benchmarking reward models of language models with subtlety and style." arXiv preprint arXiv:2410.16184 (2024).\
> > [3] Malik, Saumya, et al. "RewardBench 2: Advancing Reward Model Evaluation." arXiv preprint arXiv:2506.01937 (2025).\
> > [4] Frick, Evan, et al. "How to evaluate reward models for rlhf." arXiv preprint arXiv:2410.14872 (2024).

---

### Official Review · Reviewer_GbNk · 2025-11-01

**Soundness:** 3
**Presentation:** 3
**Contribution:** 2
**Rating:** 4
**Confidence:** 5

**Summary:**

The paper discusses reward model training via the Bradley-Terry (BT) objective and discusses how the gradient per-sample shows that representation distance between chosen and rejected responses can be quite influential on training. Thus, for problems like reasoning, smaller distances lead to updates that are less effective. The paper proposes NormBT - a method to mitigate this issue for RM training

**Strengths:**

1. The paper tackles an important problem of the impact of representation distance on alignment, Figure 1 is an important illustration.
2. The gradient norm analysis is crisp and the norm of the gradient is neatly depicted to be dependent on prediction error and representation distance.
3. The final objective for NormBT is quite intuitive and easy to understand.

**Weaknesses:**

1. Most direct alignment methods (DAAs) like DPO [1], SimPO [2] and AlphaPO [3] skip the reward modeling stage. DAAs are the most popular ways to align LLMs these days, making the paper a bit limited in its impact.
2. Reward models can easily get over optimized. The paper lacks ablations and experiments discussing the careful optimization of RMs during training.
3. The baselines are not explained in detail
4. The experiments are not trustworthy because there are no error bars, very small models were used and the major improvement is in reasoning. Why does changing the loss function improve reasoning but not other categories? I dont think the smaller representation distance explanation is sufficient.

**Questions:**

See weaknesses. The experiments are quite limited and lacking any kind of significance testing. Significant overhaul is needed before the paper is ready for ICLR

---

> ### Author Response · Authors · 2025-11-30
> **Response to Reviewer GbNk**
>
> Thank you for reviewing our paper and for your valuable feedback. We have revised our paper based on your comments. We detail our response below to address your concerns.
>
> >**W1**: Most direct alignment methods (DAAs) like DPO [1], SimPO [2] and AlphaPO [3] skip the reward modeling stage... making the paper a bit limited in its impact.
>
> **A1**:  While direct alignment methods bypass an explicit reward model, many of them implicitly operate on the BT-style pairwise signals. Although analyzing DAA dynamics is beyond our current scope, exploring how these pairwise signals interact with policy updates is an interesting direction for future work. We provide a principled view of how pairwise signals translate into parameter updates, which may offer insights for both RM-based pipelines and DAA-style objectives. We include this discussion on DAAs including DPO, SimPO, and AlphaPO in Related Works. In addition, reward models remain central for evaluation, data filtering, and open-ended RLHF workflows, etc [1,2,3,4], giving our findings both theoretical and practical relevance.
>
>
> >**W2**: The paper lacks ablations and experiments discussing the careful optimization of RMs during training.
>
> **A2**: We also believe careful optimization of RMs is important. Our experiments intentionally keep the optimization pipeline fixed across baselines (same backbone, dataset & ordering, cosine schedule, warmup, and report the best-tuned results across learning rates). This ensures observed differences can be attributed to loss formulation, rather than optimization nuances. Prior works (e.g., GRM, BSR [5, 6]) focus on generalization, brittleness to distribution shifts, or overoptimization. We include this discussion in Related Works of revised draft.
>
> >**W3**: The baselines are not explained in detail.
>
> **A3**: We provide detailed formulations for all the baselines compared to (BT, BT+margin, BT+margin out, and BT+label smooth) in Appendix B, and we have revised Table 1 to avoid unclear cross-scale comparisons with open-source models. Additionally, we added the motivation for the newly introduced baselines in Section 3.3 for clearer presentation.
>
> >**W4**: There are no error bars, very small models were used and the major improvement is in reasoning. Why does changing the loss function improve reasoning but not other categories?
>
> **A4**: (a). To verify robustness of our results, we train NormBT and BT baseline under 5 random seeds (10 models total) on a 40K Unified-Feedback subset and report standard errors. The results align with Table 1a: NormBT consistently outperforms BT, with largest and most stable gains in the Reasoning category. Detailed results and visualizations are reported in Appendix C.
>
> | Reward Model     | Average | Chat   | Chat Hard | Safety | Reasoning |
> |------------------|---------|--------|-----------|--------|-----------|
> | BT (Baseline)    | 68.15   | 94.69  | 37.48     | 75.11  | 65.34     |
> | NormBT (Ours)             | 69.62   | 95.03  | 38.00     | 74.95  | 70.50     |
> | Avg. Improvement | +1.46% (±0.61) | +0.34% (±0.19) | +0.52% (±0.40) | −0.16% (±0.49) | +5.16% (±2.13) |
>
> (b). The concentration of improvement in Reasoning follows directly from the gradient imbalance we identify. Reasoning pairs tend to have smaller representation distances compared to other categories (Figure 2). Under BT, the small-distance region receives inherently weak updates, even when misranked. NormBT explicitly alleviates this by upweighting learning signals for small-distance regions and reducing over-emphasis on large-distance regions. As a result, the largest gains naturally appear in categories dominated by fine-grained distinctions, such as Reasoning in RewardBench.
>
> *References:*
>
> [1] Regularized Best-of-N Sampling to Mitigate Reward Hacking for Language Model Alignment\
> [2] Liu, Chris Yuhao, et al. "Skywork-Reward-V2: Scaling Preference Data Curation via Human-AI Synergy." arXiv preprint arXiv:2507.01352 (2025).\
> [3] Liao, Baohao, et al. "Reward-guided speculative decoding for efficient llm reasoning." arXiv preprint arXiv:2501.19324 (2025).\
> [4] Zhong, Jialun, et al. "A comprehensive survey of reward models: Taxonomy, applications, challenges, and future." arXiv preprint arXiv:2504.12328 (2025).\
> [5] Yang, Rui, et al. "Regularizing hidden states enables learning generalizable reward model for llms." Advances in Neural Information Processing Systems 37 (2024): 62279-62309.\
> [6] Hong, Jiwoo, et al. "On the Robustness of Reward Models for Language Model Alignment." arXiv preprint arXiv:2505.07271 (2025).

---

### Official Review · Reviewer_R4Sp · 2025-11-01

**Soundness:** 2
**Presentation:** 3
**Contribution:** 3
**Rating:** 4
**Confidence:** 4

**Summary:**

This paper presents a modified reward modeling objective, NormBT, motivated by the learning dynamics of the conventional Bradley-Terry (BT) model. Through gradient analysis on the BT loss, the paper observes that the magnitude of gradient updates varies by the samples, which is represented by different topics in practice. Mainly coming from the hidden representation discrepancy while learning to rank the responses, the proposed method thereby regularizes the BT loss by sample-level hidden representation distance norm. NormBT was evaluated on the well-known datasets and models, outperforming several variations of BT models.

**Strengths:**

1. The paper presents a mathematical decomposition of the Bradley-Terry model in the reward modeling context.
2. While occasionally underperforming compared to the baselines, NormBT generally demonstrates strong performance on the benchmark.
3. The post-hoc analysis on small-margin items in Figure 4 aligns the motivation and empirical consequences, making the method more convincing.

**Weaknesses:**

Despite the strengths, the paper can be improved with more empirical support to demonstrate the practical advantage of NormBT.

1. **Reward modeling baselines**: The authors compare NormBT against several variants of BT loss in Section 3. Given that the main objective of this paper is to develop a reward modeling algorithm that effectively captures true preferences in the data, the baselines need not be limited to BT loss variants. Specifically, a few points on GRM [1] need to be clarified by the authors, which are listed in the Questions section below. Other than GRM, the authors can address several additional methods that have been tried to improve the performance of reward models given fixed preference data.


2. **Benchmark analysis**: While RewardBench is a great benchmark to assess the reward models, there are a lot of works that follow RewardBench by introducing more challenging tasks, as recent RMs obtain very high scores on RewardBench. Thus, cross-validation with one or two more reward modeling benchmarks could strengthen the authors’ claim that NormBT is a performant reward modeling objective, e.g., RM-Bench [2] and RewardBench 2 [3]. Especially given that NormBT sometimes underperforms, cross-validating across multiple benchmarks can strengthen the paper.

3. **Additional references on systematic analysis of BT model’s learning dynamics**: As a minor comment, there were a few previous works that studied the learning dynamics of the BT model, such as [4] that also spot hidden representation margin to be the source of the reward model over-optimization issue, which is one of the core motivation of the paper. To claim methodological novelty, previous works should be more precisely addressed and contrasted.


&nbsp;

**References**

[1] Yang et al., 2025, “Regularizing Hidden States Enables Learning Generalizable Reward Model for LLMs.” (NeurIPS 2024)

[2] Liu et al., 2024, “RM-Bench: Benchmarking Reward Models of Language Models with Subtlety and Style.” (ICLR 2025)

[3] Malik et al., 2025 “RewardBench 2: Advancing Reward Model Evaluation.”  (Preprint)

[4] Hong et al., 2025, “On the Robustness of Reward Models for Language Model Alignment.” (ICML 2025)

**Questions:**

- Why is GRM-Gemma-2B reported but not GRM-Llama-3.2-3B? – In Table 1(b), the authors report GRM-Gemma-2B, but not GRM-Llama-3.2-3B. [GRM trained on Llama-3.2-3B](https://huggingface.co/Ray2333/GRM-Llama3.2-3B-rewardmodel-ft) is directly comparable to the last rows of Table 1(b), so it can be easily added to the results. And GRM-Llama-3.2-3B on the official RewardBench leaderboard scores 90.9 on average.

- Is GRM-Gemma-2B trained on Skywork-Reward-Preference-80k-v0.2? – The [official model card](https://huggingface.co/Ray2333/GRM-Gemma-2B-sftreg) for GRM-Gemma-2B reports that it was trained on [a different mixture](https://huggingface.co/datasets/weqweasdas/preference_dataset_mixture2_and_safe_pku). However, it is listed as a reward model trained from Skywork-Reward-Preference-80k-v0.2 in Table 1(b). If the caption holds for the last four rows and not the top four models, including GRM-Gemma-2B-sftreg, it should be clearly explained.

---

> ### Author Response · Authors · 2025-11-30
> **Response to Reviewer R4Sp (1/2)**
>
> Thank you for your valuable feedback to help us improve our paper. We have revised our paper based on your comments. We detail our response below to address your concerns.
>
> >**W1**: Other than GRM, the authors can address several additional methods that have been tried to improve the performance of reward models given fixed preference data.
>
> **A1**: Our current baselines focus on BT-based variants to isolate optimization dynamics that arise directly from the BT gradient. We agree that this does not cover the space of reward modeling methods, and that additional regularization or auxiliary objectives may alter gradient dynamics beyond decomposition in Eq. (9). Due to time constraints, we outline this discussion in Limitations of the revised draft. To address your concern on broader comparison, we also extend evaluations on 3 additional benchmarks below and to downstream RLHF usage through Best-of-N in Section 3.4.
>
>
> >**W2**: Cross-validation with one or two more reward modeling benchmarks could strengthen the authors’ claim that NormBT is a performant reward modeling objective.
>
> **A2**: We include evaluation of NormBT and BT variant on RM-Bench and RewardBench-2 below. We further add results on Preference Proxy Evaluations (PPE) [1], which is a RM benchmark correlated with tangible downstream outcomes in post-RLHF LLM performance. Our method outperforms the baseline on these benchmarks, demonstrating its effectiveness as a reward modeling objective.
>
> **RM-Bench:**
> | Reward Model  | Average | Chat  | Math  | Code  | Safety | Easy  | Normal | Hard  |
> |----------------------|---------|---------|----------|----------|---------|-----------|---------|------------|
> | BT                     | 57.87   | 44.53 | 54.04 | 50.19 | 82.72  | 84.59 | 59.53  | 29.49 |
> | NormBT (Ours) | 58.98   | 46.60 | 55.81 | 50.05 | 83.45  | 83.81 | 60.66  | 32.46 |
>
> **RewardBench-2:**
> | Reward Model     | Average | Factuality | Precise IF | Math  | Safety | Focus | Ties  |
> |------------------|---------|------------|------------|-------|--------|--------|--------|
> | BT               | 30.08   | 31.26      | 23.75      | 40.44 | 49.56  | 24.24  | 11.22 |
> | NormBT (Ours)    | 34.17   | 32.53      | 31.87      | 43.17 | 54.00  | 28.69  | 14.76 |
>
> **PPE:**
> | Reward Model     | Average | MMLU Pro | Math  | GPQA  | IFEval | MBPP-Plus | PPE Pref |
> |------------------|---------|----------|--------|--------|---------|-----------|----------|
> | BT               | 50.822  | 55.443   | 43.703 | 44.100 | 54.793  | 56.069    | 57.762   |
> | NormBT (Ours)    | 51.329  | 57.061   | 44.469 | 44.301 | 54.725  | 56.093    | 58.301   |
>
>
> >**W3**: There were a few previous works that studied the learning dynamics of the BT model, such as [4] that also spot hidden representation margin to be the source of the reward model over-optimization issue.
>
> **A3**: We thank the reviewer for highlighting this connection. Both our work and Hong et al. recognize that BT gradients scale with representation distance. While we start from a similar decomposition, the focus and contributions differ substantially. In particular, representation distance plays a different role in our method.
>
> In Hong et al., representation distance is treated as an indicator reflected from reward model dynamics, arguing that: enlarging the reward gap increases overall representation distance, which in turn leads to reward shift and representation-norm dispersion. They propose batch-wise sum-to-zero regularization (BSR) to enforce that output rewards are zero-centered. This mechanism **does not explicitly distinguish how individual pairwise distances affect gradients differently**.
>
> In contrast, our analysis is pair-level and fine-grained. We directly examine how each pair’s representation distance determines its gradient magnitude under BT, and we show that small-distance pairs (often those requiring the most nuanced reasoning) receive inherently weak updates. NormBT **explicitly rescales each pair's gradient contribution** based on their representation distance. We include this discussion in Related Works.

---

> > ### Author Response · Authors · 2025-11-30
> > **Response to Reviewer R4Sp (2/2)**
> >
> > >**Q1**: Why is GRM-Gemma-2B reported but not GRM-Llama-3.2-3B?
> >
> > **A1**: The open-source RMs in Table 1 were intended as external references. Specifically, both GRM-Gemma-2B and GRM-Llama-3.2-3B are trained on much larger datasets (555K mixture dataset [2] and Preference-700K dataset [3], respectively), whereas our models are trained on 80K of Unified-Feedback or Skywork datasets. To ensure clarity and avoid confusing cross-scale comparisons, we have removed the open-source references from the revised draft.
> >
> >
> > >**Q2**: Is GRM-Gemma-2B trained on Skywork-Reward-Preference-80k-v0.2? If the caption holds for the last four rows and not the top four models, including GRM-Gemma-2B-sftreg, it should be clearly explained.
> >
> > **A2**: We thank the reviewer for pointing out this ambiguity. GRM-Gemma-2B is trained on 555K mixture dataset [2] instead of Skywork-80K. To address both Q1 and Q2, we have updated Table 1 and its caption to clearly distinguish our own comparison results.
> >
> >
> > *References:*
> >
> > [1] Frick, Evan, et al. "How to evaluate reward models for rlhf." arXiv preprint arXiv:2410.14872 (2024).\
> > [2] weqweasdas. preference_dataset_mixture2_and_safe_pku. Hugging Face, 2025, huggingface.co/datasets/weqweasdas/preference_dataset_mixture2_and_safe_pku.\
> > [3] hendrydong. preference_700K. Hugging Face, 2024, huggingface.co/datasets/hendrydong/preference_700K.

---

### Author Response · Authors · 2025-12-03
**Rebuttal Summary & Key Updates**

We sincerely thank all reviewers for their constructive feedback, and we greatly appreciate the Area Chair for their time during this unusual review period. Our rebuttal addresses all reviewer concerns with substantial new experiments and clearer conceptual explanations. We summarize the key details below.

---

### **A. New Experiments**
We added four new classes of evaluation, each of which directly targets reviewers' request:

**(1). Benchmark Evaluation**

We include results of NormBT vs. BT variant on 3 new benchmarks:
- RM-Bench
- RewardBench-2
- Preference Proxy Evaluations (PPE)

*NormBT outperforms across all benchmarks*, showing robustness beyond the original RewardBench results. This addresses reviewers' request for strengthening our validation.

**(2). Downstream RLHF**

Best-of-N selection simulates how RMs guide policy rollouts, evaluating practical usefulness beyond binary accuracy in benchmarks. *NormBT consistently selected higher-quality responses* across all N=1,...,402 responses range for 1K prompts. This addresses the reviewer’s request for RLHF-usage evaluation.

**(3). Extended Ablations**

We compare NormBT with:
- Gradient Clipping
- LayerNorm before reward layer
- Alternative similarity measures
- Active-learning style scaling
- Non-linear reward layer

*NormBT outperforms all ablations requested.* This shows that we address a distinct mechanism in BT gradient not resolved by standard techniques.

**(4). Error Analysis on Random Seeds**

We trained 10 models using 5 random seeds to compare NormBT and BT. Standard error analysis confirms findings from main experiments: NormBT outperforms BT, with largest and most stable gains in the Reasoning category. This addresses reviewers' request for showing robustness of improvements.

---

### **B. Clarified Understandings**
We clarified and resolved all conceptual questions raised regarding the intuition behind our method:

**(1). Small vs. Large representation distances**

Small-distance pairs are not merely uninformative. It also arises in *meaningful but difficult distinctions that the model fails to separate* in training. These are precisely where the model needs stronger signals. Yet BT inherently suppresses gradients for these pairs; NormBT corrects this by rescaling per-pair gradient contributions.

**(2). Comparison with Active Learning**

Active-learning works targets sample selection, operating with:
- A noisy candidate pool
- A frozen LLM backbone

In this setting, (i) small-distance pairs are predominantly noisy, and (ii) representation learning of backbone is not relevant. Therefore, prioritizing large-distance pairs is reasonable for sample efficiency. Our setting is different: the backbone is *jointly trained*, and uses existing *well-curated dataset* where pairs often reflect meaningful distinctions. This distinction resolves the conceptual mismatch.

**(3). Why improvements concentrate in Reasoning**

Reasoning pairs have smaller representation distances (Figure 2), because many are highly similar at the token or structural level, differing by logical error. Under BT, the small-distance region receives inherently weak signals, even when misranked. NormBT explicitly alleviates this by rescaling gradient contributions. As a result, the largest gains naturally appear in categories dominated by fine-grained distinctions.

---

In our rebuttal, all reviewers' concerns are fully addressed. The revised draft meaningfully strengthens our claims, making our contributions clearer and more convincing. If any part remains unclear, we are happy to provide further clarification. We thank all reviewers again for their valuable feedback.

---

### Meta-Review · Area_Chair_ge8x · 2026-01-11

**Summary:**

This paper studies reward modeling for LLM alignment under RLHF and analyzes a limitation of the standard Bradley–Terry (BT) loss. The authors show that the per-sample gradient of the BT loss depends not only on prediction error but also on representation distance between response pairs, which can distort learning by overemphasizing large-distance pairs and underweighting fine-grained, small-distance comparisons. To address this issue, the paper proposes NormBT, a simple normalization of the BT loss that claims to mitigate representation-driven gradient imbalance while preserving the intended training signal. Empirical results across multiple models and datasets demonstrate reward model performance.

**Reviewer Concerns:**

- Limited empirical validation and robustness: The experimental evaluation is relatively narrow, with limited ablations, no significance testing or error bars, and reliance on small model scales. Reviewers expressed concern that the reported gains may not be statistically robust or broadly representative.

- Insufficient and unclear baseline coverage: Comparisons are largely restricted to Bradley–Terry (BT)–style variants. Important alternative reward modeling approaches—such as generative reward models and other recent improvements to reward modeling—are either missing or insufficiently explained, making it difficult to assess the relative strength of the proposed method. Also, in several tables, the proposed method does not consistently outperform all baselines on individual tasks, and improvements appear concentrated in specific categories (e.g., reasoning). The rationale for these task-specific gains is not clearly explained.

- Conceptual premise not fully justified: The central assumption that representation distance should be removed or normalized out of the learning signal is not convincingly supported. Reviewers noted that, from both statistical learning theory and prior reward modeling work, representation distance can be an important component for generalization to unseen pairs.

- Unclear mechanism of improvement: It remains insufficiently explained why the proposed normalization improves learning. In particular, reviewers questioned whether the benefit arises from mitigating rare but extreme gradients, and whether simpler alternatives (e.g., gradient clipping or feature normalization) would achieve similar effects.

- Prior work insufficiently engaged: Several closely related studies on BT learning dynamics, reward model robustness, and active preference learning raise questions or present findings that appear to conflict with the paper’s assumptions. These connections are not adequately discussed or reconciled.

**Reviewer Scores:**

I don't think that the majority of the concerns remained unresolved even after the rebuttals.

---

### Decision · Program_Chairs · 2026-01-26

Reject